# Conformational coupling of the sialic acid TRAP transporter HiSiaQM with its substrate binding protein HiSiaP

Martin F. Peter [1,4,5], Jan A. Ruland [2,5], Yeojin Kim [1,5], Philipp Hendricks [1,5], Niels Schneberger [1], Jan Peter Siebrasse[2], Gavin H. Thomas [3], Ulrich Kubitscheck [2] & Gregor Hagelueken [1] ✉

The tripartite ATP-independent periplasmic (TRAP) transporters use an extra cytoplasmic substrate binding protein (SBP) to transport a wide variety of substrates in bacteria and archaea. The SBP can adopt an open- or closed state depending on the presence of substrate. The two transmembrane domains of TRAP transporters form a monomeric elevator whose function is strictly dependent on the presence of a sodium ion gradient. Insights from experimental structures, structural predictions and molecular modeling have suggested a conformational coupling between the membrane elevator and the substrate binding protein. Here, we use a disulfide engineering approach to lock the TRAP transporter HiSiaPQM from *Haemophilus influenzae* in different conformational states. The SBP, HiSiaP, is locked in its substrate-bound form and the transmembrane elevator, HiSiaQM, is locked in either its assumed inward- or outward-facing states. We characterize the disulfide-locked constructs and use single-molecule total internal reflection fluorescence (TIRF) microscopy to study their interactions. Our experiments demonstrate that the SBP and the transmembrane elevator are indeed conformationally coupled, meaning that the open and closed state of the SBP recognize specific conformational states of the transporter and vice versa.

Tripartite ATP-independent periplasmic (TRAP) transporters are widely distributed in bacteria and archaea, and are also found in common pathogens[1,2]. Together with the ATP-binding cassette (ABC) importers and the tripartite tricarboxylate transporters (TTT), they define the three classes of substrate binding protein (SBP)-dependent transporters. The SBP (commonly referred to as the P-domain in TRAP transporters) is a soluble protein that freely diffuses in the periplasm of Gram-negative bacteria or is associated with the cell membrane in Gram-positive bacteria and archaea[3]. It scavenges substrate molecules and delivers them to the membrane transporter. It is believed that the SBP is advantageous in situations where low substrate concentrations are encountered and that it serves as a substrate store[4].

The first structural information about TRAP transporters was provided by two high-resolution crystal structures of the P-domain of the sialic acid (Neu5Ac, N-Acetylneuraminic acid) TRAP transporter HiSiaPQM from *Haemophilus influenzae* in both the apo and holo states (i.e., substrate-bound and -free, respectively)[5,6]. Since then, many more structures have been determined, confirming the general architecture of the P-domain as two globular lobes connected by a long backbone helix (Fig. 1, red). Substrate binding induces a bend in this

[1]Institute of Structural Biology, University of Bonn, Venusberg-Campus 1, 53127 Bonn, Germany. [2]Clausius Institute for Physical and Theoretical Chemistry, University of Bonn, Wegelerstr. 12, 53115 Bonn, Germany. [3]Department of Biology (Area 10), University of York, York YO10 5YW, UK. [4]Present address: Biochemistry Center, Heidelberg University, Im Neuenheimer Feld 328, 69120 Heidelberg, Germany. [5]These authors contributed equally: Martin F. Peter, Jan A. Ruland, Yeojin Kim, Philipp Hendricks. ✉e-mail: hagelueken@uni-bonn.de

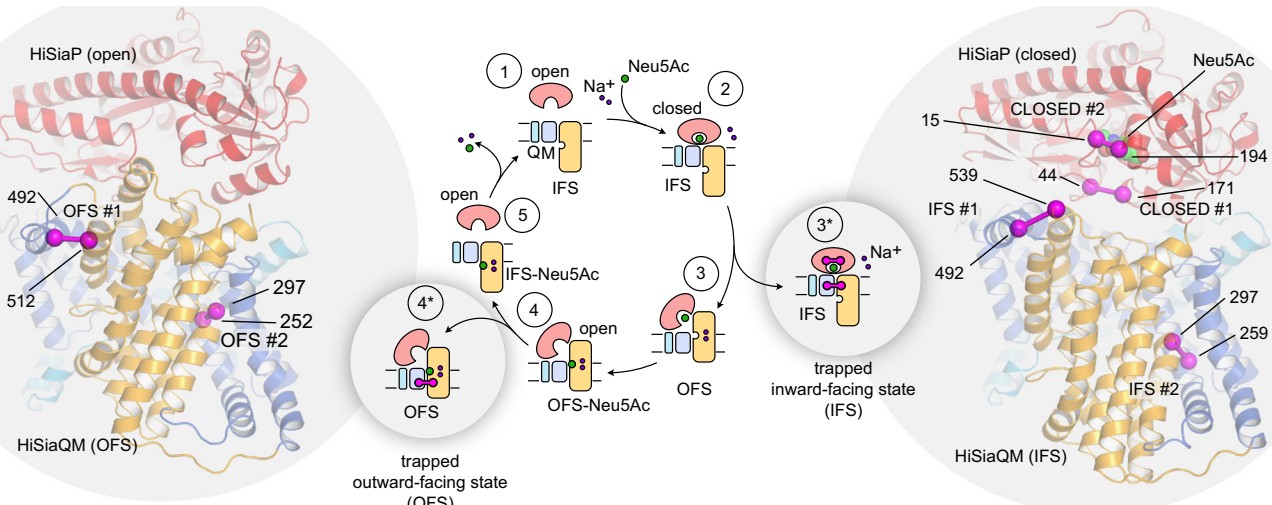

**Fig. 1 | Trapping a TRAP transporter.** The transport cycle of TRAP transporters as described in[14] is depicted schematically in the middle of the figure. To trap the transporter in specific states, the stator and elevator domains were cross-linked by disulfide engineering to yield states 2* and 4*. The cartoon representations of HiSiaPQM on the left and right of the figure are AlphaFold2 models of the tripartite complex[14]. The individual domains of the transporter are colored as in the schematic (HiSiaP: red, HiSiaQM stator: blue, HiSiaQM elevator: yellow). Residues that were mutated to cysteines to form disulfide bonds are indicated by magenta spheres and the disulfide bond is shown as a magenta bar connecting the two residues. In state 2*, the bound Neu5Ac is shown as spheres. IFS: inward-facing state; OFS: outward-facing state; Neu5Ac: N-Acetylneuraminic acid.

helix, closing the two lobes around the substrate molecule. This motion is often compared to that of a Venus flytrap and has been studied in detail, not only with high-resolution structures, but also with biophysical techniques and molecular dynamics (MD) simulations[7–12]. In contrast, structural information on the two transmembrane domains, i.e., the smaller Q-domain with 4 transmembrane (TM) helices and the larger M-domain with 11 TM helices (Fig. 1 blue, yellow), has only recently been obtained. Two cryo-EM structures revealed that TRAP transporters are elevator-type transporters (reviewed in[13]), but with a striking twist: TRAP transporters are monomeric elevators[14,15]. The key to this unusual feature is the Q-domain, the function of which has been a mystery for a long time. The structures revealed that its four TM helices form a unique helical sheet that wraps around the M-domain and serves to enlarge the stator portion (also known as scaffold- or oligomerization-domain) of the latter. This is thought to anchor the stator domain in the membrane, allowing the elevator portion to move up and down, yielding the outward- and inward-facing states of the transporter (OFS and IFS, respectively). The anchoring role of the Q-domain is supported by strongly bound annular lipids that are visible in a recent higher resolution cryo-EM structure of HiSiaQM[16].

How the conformationally flexible SBP interacts with these different states of the moving elevator transporter is an interesting structural question. Initial insights have come from a combination of experimental structures, structural predictions using the AlphaFold2 algorithm and repeat swap modeling[14,17,18]. It was predicted that the closed-state SBP forms a complex with the IFS of the elevator and that the open-state SBP structurally matches and interacts with its OFS. While the predicted binding interfaces could be confirmed by mutagenesis[14,15], a direct observation of this conformational coupling would support the prediction and significantly improve our understanding of TRAP transporters.

Here, we use disulfide engineering techniques[19,20] to trap the three domains of the sialic acid TRAP transporter HiSiaPQM in its major conformational states, i.e., a permanently closed state of the P-domain, as well as the IFS and OFS of the elevator domains. We use X-ray crystallography, biochemical- and biophysical approaches to characterize the conformationally trapped domains. We then combine the different trapped domains and use single molecule total internal reflection fluorescence (TIRF) microscopy to study their interactions.

## Results

### Design of a conformationally trapped TRAP transporter

We searched the structural model of the HiSiaPQM complex for sites where the different domains can be locked in a given conformational state by intra-chain disulfide bonds. As shown in Fig. 1 (right), we selected two pairs of residues in HiSiaP (S44C/S171C and S15C/A194C) and named the two constructs HiSiaP CLOSED #1 and HiSiaP CLOSED #2, respectively. For both constructs, the thiol groups of the introduced cysteines would likely be in close proximity in the substrate-bound state (Supplementary Fig. 1, Supplementary Table 1), and the formation of a disulfide bond would therefore be expected to lock the SBP in this conformation. We and others have previously shown with various techniques that HiSiaP is stabilized in its open conformation in the absence of sialic acid[9–11,8], allowing us to use the apo protein in experiments addressing the open state of HiSiaP.

For HiSiaQM, we selected four pairs of residues to be converted to disulfides. The pairs A492C/Q539C and M259C/M297C should lock the elevator domain in its IFS (HiSiaQM IFS #1 and HiSiaQM IFS #2, respectively), which is probably the resting state of the protein as it has now been observed in three experimental structures[14–16], while the pairs A492C/L512C and Y252C/M297C would trap the OFS (HiSiaQM OFS #1 and HiSiaQM OFS #2, respectively) (Fig. 1, left, Supplementary Fig. 1). The selection was guided by our previously published structural models[14] and inspired by the residues selected by Mulligan et al.[20] to trap the structurally related VcINDY elevator transporter in its IFS and OFS. While such crosslinking experiments are very helpful to uncover mechanistically important conformational changes, the presence of a small fraction of non-crosslinked molecules is hard to exclude and should be considered during the interpretation of the results.

### Disulfide linked HiSiaP mutants bind Neu5Ac and can be trapped in the closed state

The CLOSED #1 and #2 constructs of HiSiaP (Fig. 1) were expressed in *E. coli* and were purified analogously to the wild-type protein, which does not have any native cysteines. The proteins behaved like the wild type in gelfiltration runs (Supplementary Fig. 2). To confirm that the double

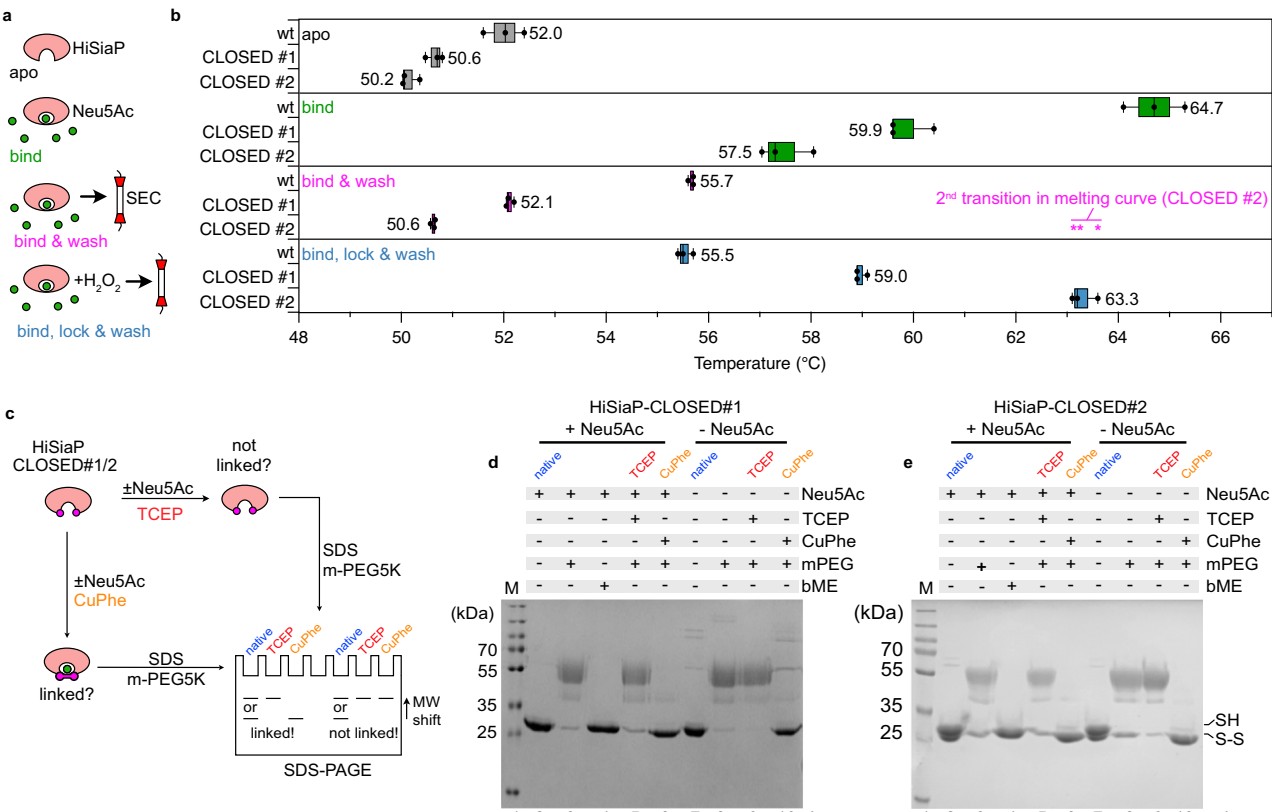

**Fig. 2 | Trapping HiSiaP in its Neu5Ac-bound closed state. a** Schematic illustrating the sample preparation for the nanoDSF experiments in (**b**). **b** Melting points (nanoDSF) of apo HiSiaP (dark gray), HiSiaP in the presence of 1 mM Neu5Ac (green, bind), the HiSiaP/Neu5Ac complex following ligand removal by gel filtration (magenta, bind & wash) and the $H_2O_2$- and then gel filtration-treated sample (blue, bind, lock & wash) for wildtype, CLOSED #1 and CLOSED #2 HiSiaP. The raw data are shown in Supplementary Fig. 4. The asterisks indicate the position of a distinct 2nd peak in the 1st derivative of the bind & wash melting curve for the CLOSED #2 construct. The asterisk is also present in Supplementary Fig. 4. Each box represents $n = 3$ independent experiments (shown as circles). Here, the minimum and maximum are shown as whiskers, the median is shown as a vertical line and the left and right boundaries of the box represents quartiles. **c** Schematic explaining the PEGylation (using a PEG5K malemide (mPEG5K)) assay to quantify the formation of disulfide bonds. **d** PEGylation assay of the CLOSED #1 construct. The experiment was performed two times ($n = 2$) with similar results. **e** PEGylation assay of the CLOSED #2 construct. The experiment was performed three times ($n = 3$) with similar results. Uncropped versions of the gels in (**d**) and (**e**) are shown in Supplementary Fig. 10. Neu5Ac N-Acetylneuraminic acid, PEG polyethylene glycol, DSF differential scanning fluorimetry, bME ß-mercaptoethanol. Source data are provided as a Source Data file.

cysteine mutants were still able to bind Neu5Ac, we used isothermal titration calorimetry (ITC) and determined an average dissociation constant ($K_D$) of 61 nM for the wild-type HiSiaP protein (Supplementary Fig. 3a–c), in agreement with previous studies[12,14]. HiSiaP CLOSED #1 clearly bound sialic acid, but with an approximately 13-fold weaker average affinity ($K_D = 0.8\,\mu M$, Supplementary Fig. 3d–f). It is known that even very conservative changes of the residues in the Neu5Ac binding pocket can lead to quite dramatic changes in the $K_D$[7].

The second construct, HiSiaP CLOSED #2, showed a peculiar behavior in the ITC experiments that was distinct from any other HiSiaP construct that we had previously tested. The titration curves of HiSiaP CLOSED #2 consistently had a biphasic shape and could not be fitted with a simple 1:1 binding model (Supplementary Fig. 3g–i). Instead, a model with two independent binding reactions was used. One of the reactions was fully resolved and the average $K_D$ was determined to be in the micromolar range (~2 µM). The other reaction appeared to have a higher affinity but was not fully resolved in our titration experiments and hence its thermodynamic parameters could not be reliably determined. We speculate that this observation was due to a fraction of HiSiaP CLOSED #2 having formed its engineered disulfide bond. Overall, our ITC data suggested that the presence of the cysteines alone, in the absence of an oxidizing agent, results in only a small fraction of the HiSiaP with the desired disulfide bonds formed. To investigate this further and to check if the protein was indeed

trapped in the closed state, we used differential scanning fluorometry (nanoDSF) and analyzed the stability of the HiSiaP constructs in the conditions summarized in Fig. 2a.

As expected, binding of sialic acid to wild-type HiSiaP resulted in a significant thermal stabilization of the protein with a ~13 °C shift in its melting temperature from 52 °C to 64.7 °C (Fig. 2b apo and bind, Supplementary Fig. 4b–d). For the HiSiaP CLOSED #1 and #2 constructs, the addition of sialic acid also clearly stabilized the proteins (50.6 °C -> 59.9 °C and 50.2 °C-> 57.5 °C, respectively), but not as much as observed for the wild type, probably reflecting the lower affinity of the constructs for Neu5Ac. In the next set of experiments, we incubated the proteins with sialic acid and then removed the excess sialic acid by gel filtration (Fig. 2a, b bind & wash). Here, the wild-type protein had only a slightly elevated melting temperature of 55.7 °C, since the bound sialic acid was gradually lost during the gel filtration experiment. We found distinctly biphasic melting curves for the HiSiaP CLOSED #1 and #2 constructs (Supplementary Fig. 4e–j), where the largest fraction had melting temperatures very similar to the samples without any Neu5Ac added (Fig. 2b). Again, this is in line with the ITC experiments, where both mutants showed a lower substrate affinity compared to the wildtype and due to this more easily release the substrate again. The smaller fraction, however, had an increased melting temperature, which was particularly evident for HiSiaP CLOSED #2 (magenta asterisks in Fig. 2b and Supplementary Fig. 4).

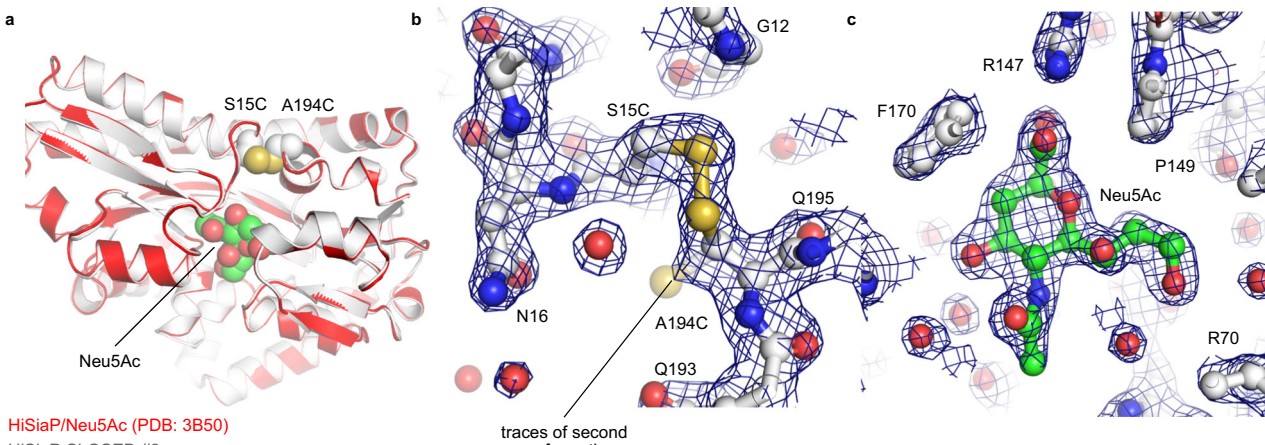

**Fig. 3 | Crystal structure of the HiSiaP CLOSED #2 construct. a** Crystal structure of HiSiaP 15 C/194 C (CLOSED #2) superposed onto the closed-state crystal structure of HiSiaP (red) in complex with Neu5Ac (PDB-ID: 3B50[6]). The two introduced cysteines at positions 15 and 194 are shown as spheres. **b** Detail of the disulfide bond with 2mF$_o$-DF$_c$ density contoured at 1.5 σ. The map shows that a fraction of the cysteine at position 194 populates a different rotamer and is hence not involved in a disulfide bridge. The positions of selected residues are indicated. **c** 2mF$_o$-DF$_c$ density map (contoured at 1.5 σ) of the bound Neu5Ac molecule. The locations of selected residues are indicated. Neu5Ac: N-Acetylneuraminic acid.

This was reminiscent of the peculiar observation in the ITC experiments (Supplementary Fig. 3) and supports the notion that a portion of HiSiaP CLOSED #2 had a closed disulfide bond after substrate addition. To increase this fraction, we added H$_2$O$_2$ to the Neu5Ac binding step of the experiment and removed the oxidant together with the unbound Neu5Ac by gel filtration (Fig. 2b, bind, lock & wash). Strikingly, the nanoDSF analysis of these samples showed that the melting point of the two constructs was now significantly higher than that of the apo form and coincided with the distinct shoulder observed in the bind & wash experiment (magenta asterisks in Fig. 2b). Judged by the relative heights of the peaks in the nanoDSF curves (Supplementary Fig. 4), we estimate that >90 % of the SBPs had been successfully locked in their closed state. This was confirmed by a PEGylation assay[20] as sketched in Fig. 2c. Lanes 2–6 on the corresponding SDS-PAGE gels for CLOSED #1/2 (Fig. 2d, e) showed the expected pattern (see Fig. 2c) for nearly quantitative disulfide formation.

### Engineered disulfide links reveal rare closure events of apo HiSiaP

According to the proposed TRAP transporter mechanism (Fig. 1, ref. 3,14), the state of the SBP in solution is an important factor in the transport cycle. Single molecule FRET experiments had shown that the substrate-bound P-domain reopens after ~125 ms[8]. In contrast, spontaneous closure in the absence of Neu5Ac was not observed experimentally, neither with single molecule FRET nor with pulsed electron paramagnetic resonance (EPR) experiments or X-ray crystallography[8–10]. Thus, if at all, such short-lived events must happen quite rarely and are hence difficult to detect. On the other hand, molecular dynamics (MD) simulations of the closely related VcSiaP from *Vibrio cholerae* suggested that semi-closed states of the SBP can occur, even in the absence of Neu5Ac[11].

Inspired by a referees' comment, we investigated, whether the disulfide-engineered constructs designed in this work can be used as a tool to detect such rare events. In principle, if a particular HiSiaP molecule briefly visits a conformational state that fulfills the geometric requirements for disulfide bond formation, the crosslink should form with a high probability, at least in the presence of an oxidation agent. Given enough time, the growing pool of crosslinked molecules should then be detectable.

Firstly, we investigated the running behavior of the freshly prepared and untreated CLOSED #1/2 mutants on non-reducing SDS-PAGE gels. While CLOSED #1 looked normal, we detected a distinct double band for CLOSED #2 (Fig. 2e, lane 2). This double band was not present on reducing gels, indicating that the observed band pattern was indeed caused by the spontaneously formed disulfide bond (Fig. 2e, lane 4). Strikingly, in the presence of the oxidation agent copper phenantroline (CuPhe) and notably in the absence of Neu5Ac, the upper band shifted completely to the lower band (Fig. 2e, lane 10). Thus, within the time of the experiment (1 h), all HiSiaP CLOSED #2 molecules in the sample had formed the disulfide link. We repeated the experiment with CLOSED #1 and also here, the disulfide bridge was formed, as detected by a PEGylation experiment (Fig. 2d, lane 10). The different position of the engineered cysteines along the edge of the substrate binding pocket explains the higher propensity of CLOSED #2 to spontaneously form a disulfide bond, even without an oxidation agent (Supplementary Fig. 5a). This also rationalizes the biphasic ITC and nanoDSF curves of the CLOSED #2 construct (see previous section).

The distinct band shift of the CLOSED #2 construct allowed us to perform a straight-forward time course experiment by determining the fractions of crosslinked and not-crosslinked molecules at different time points by SDS-PAGE analysis (Supplementary Fig. 5b). The band intensities were quantified in two independent experiments and could in each case be fitted with a mono-exponential decay function with a mean decay constant λ of ~ 0.044/s ($n = 2$). This results in a mean half-life (t$_{1/2}$) of ~ 16 s and a mean average life time τ of 23 s. Assuming that the wild type protein behaves similar to the CLOSED #1/2 constructs, we can estimate that on average, the apo HiSiaP protein performs a closing motion every ~23 s. Knowing the mean decay constant λ, we can estimate that within one millisecond less than ~0.01 % of all P-domains have performed a closing motion. In contrast to the CLOSED #1/2 constructs, the wild-type P-domains quickly opens again[11] and thus, only a small fraction of closed state P-domains is present at any time. This explains why such rare events were not detected by our previous experiments[8–10].

### A crystal structure of HiSiaP trapped in its closed state

To verify that the disulfide-linked SBPs had the expected structure, we set up crystallization experiments of the H$_2$O$_2$-treated HiSiaP CLOSED #1/2 constructs. Heavily intergrown crystals of HiSiaP CLOSED #2 were obtained and single crystalline pieces could be broken off for diffraction data collection at the PETRA III synchrotron (DESY, Hamburg, Germany). A 1.9 Å data set (space group I222) was collected and the structure was solved by molecular replacement using the closed state

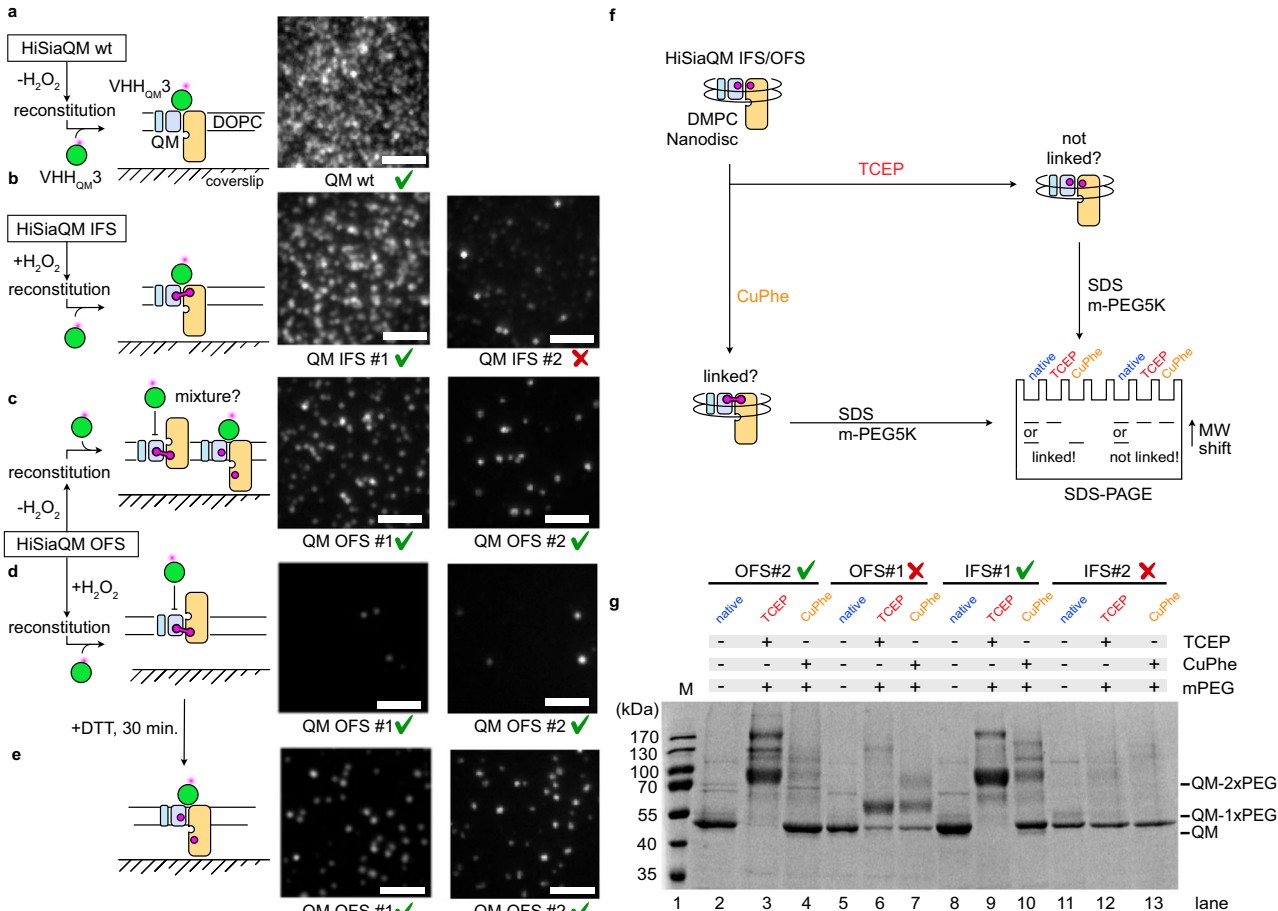

**Fig. 4 | VHH$_{QM}$3 binding to the IFS/OFS #1/2 constructs of HiSiaQM. a** The schematic on the left depicts the setup of the single molecule TIRF experiment. The micrograph on the right shows that VHH$_{QM}$3 interacts with the wild-type QM-domains in the bilayer. **b** Same as (**a**)) but with HiSiaQM IFS #1/2. **c** Same as (**a**), but with HiSiaQM OFS #1/2 and without adding H$_2$O$_2$ during the reconstitution. **d** Same as (**c**) but with adding H$_2$O$_2$ during the reconstitution. **e** Experiment in (**d**) but after the addition of DTT and a 30 min. equilibration time. The scale bars equal 3 µm. **f** Schematic explaining the PEGylation assay[20] for the HiSiaQM constructs.

**g** PEGylation experiment as described in (**f**). Throughout all panels, results that met our expectations are marked with a green checkmark. Validation results that did not meet our expectations are marked with a red cross. The experiments in (**a**–**e**) were performed three times (*n* = 3) with similar results. The experiment in (**g**) was performed three times (*n* = 3) with similar results. An uncropped version of the gel in (**g**) is shown in Supplementary Fig. 10. IFS inward-facing state, OFS outward-facing state, PEG polyethylene glycol, DTT dithiothreitol, TIRF total internal reflection fluorescence.

of HiSiaP as a search model (PDB-ID: 3B50[6]). A clear solution was found (Fig. 3a) and inspection of the electron density map revealed density at the expected position of the disulfide bond (Fig. 3b) and the sialic acid binding site (Fig. 3c), demonstrating that the disulfide bond had indeed been formed and the sialic acid molecule had been captured in its binding site. The structure was refined to R/R$_{free}$ factors of 0.211/0.259 and good geometric parameters as determined by MolProbity[21] (Supplementary Table 2). Figure 3a shows an overlay of the HiSiaP CLOSED #2 structure with that of the closed-state of the wild-type protein (PDB-ID: 3B50[6]). The two structures overlap with an r.m.s.d. value of 0.24 Å over 300 C$_α$ atoms, and except for the introduced disulfide bond, there is no significant change between the backbone or side chain atoms of the two structures. Even the positions of the visible water molecules in the binding site were very similar. We found that traces of a second conformation of C194 were visible in the electron density (Fig. 3c). This might either indicate incomplete formation of the disulfide bridge or radiation induced reduction of the latter[22].

Taken together, the experiments above (ITC, nanoDSF and PEGylation assay) showed the expected formation of disulfide bonds and trapping of the closed state of HiSiaP. Additionally, both mutants showed very comparable results. Since the structure of HiSiaP CLOSED #2 could also be experimentally verified, this mutant is used in the TIRF

experiments below. However, the equivalent TIRF experiments were also performed with CLOSED #1 leading to very similar results (see below).

## Disulfide-linking the elevator domain of HiSiaQM

We expressed and purified the HiSiaQM IFS/OFS double cysteine constructs shown in Fig. 1. To avoid any interference of the engineered cysteines with the five native cysteines of HiSiaQM, we replaced the cysteine sidechains with serine or alanine (C94A, C325S, C334S, C400S, C485S), yielding HiSiaQM$_{ΔCys}$, which is for simplicity termed HiSiaQM below. According to our established in vivo uptake assay[14,23,24], the transporter construct was still active, albeit at a reduced level compared to the wild type (Supplementary Fig. 6). In each case, the protein behaved very similar to the wild type during purification (Supplementary Fig. 2). In addition to the above described PEGylation assay, we used single-molecule TIRF microscopy and AF555-VHH$_{QM}$3, a fluorescent labeled Nanobody™[14], to probe the conformational state of the disulfide-bonded QM-domains and thereby to test whether the engineered disulfide bonds had indeed been formed. As known from our cryo-EM structure[14], VHH$_{QM}$3 binds on the periplasmic side of HiSiaQM, at the boundary between the stator domain and the inward-facing elevator domain (i.e. the IFS) with a very high affinity of

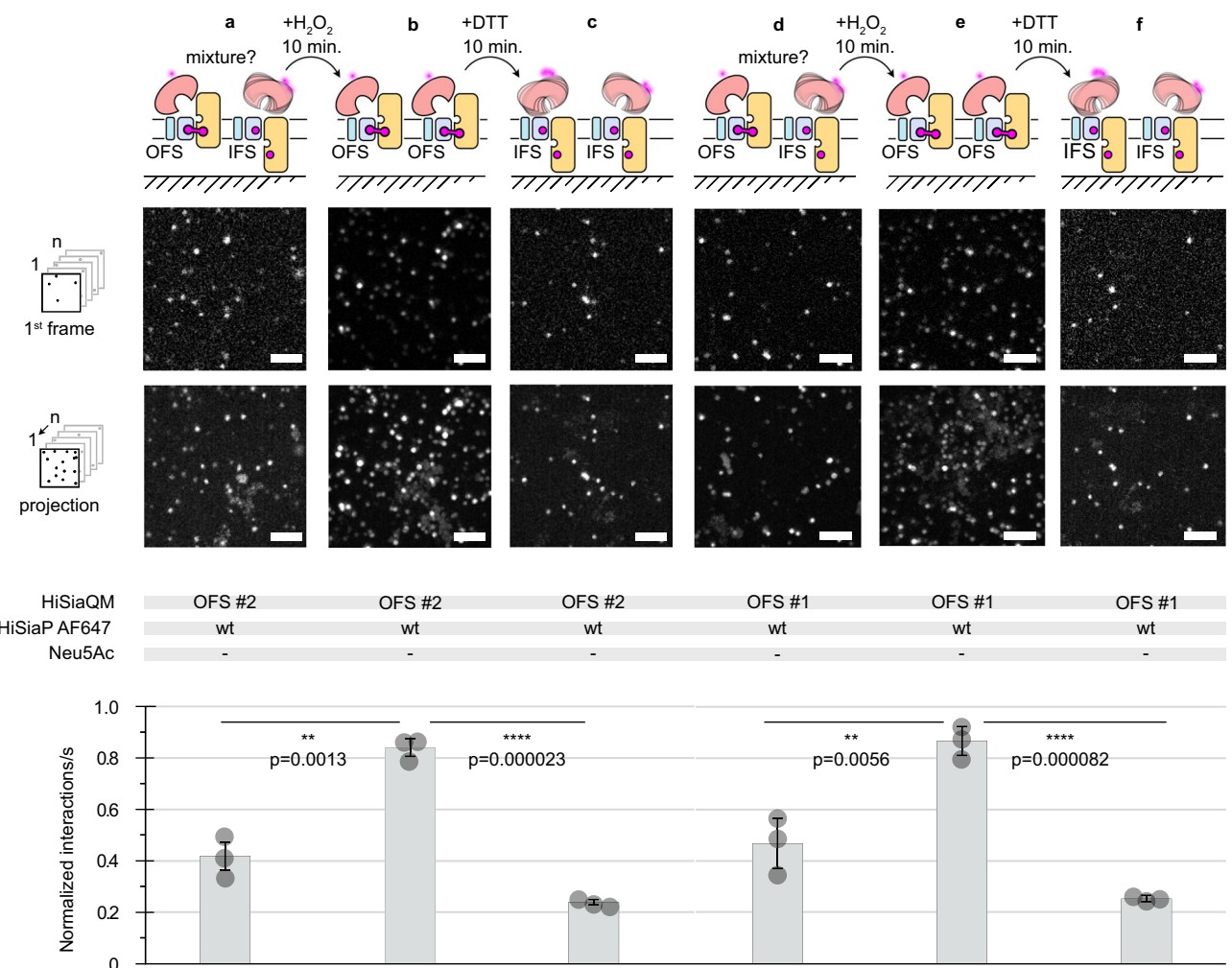

**Fig. 5 | Following disulfide formation in HiSiaQM OFS #1 in real time. a** Top: HiSiaQM OFS #2 was not pre-treated with $H_2O_2$ and incorporated into DOPC SSBs and interactions with the AF647 labeled P-domain were observed in the absence of sialic acid. The first frames of a typical image sequence with a length of 1000 frames and the corresponding maximum intensity projection of the complete image sequence is shown. Each frame was integrated for 10 ms. Scale bars, 3 μm. Bottom: Quantification of the imaging data: The statistical significance of differences between selected experiments was assessed by applying a two-sided unpaired Student's $t$ test with a 95% confidence interval. (*$p < 0.01$, **$p < 0.001$). The full dots represent the average results of $n = 3$ independently prepared samples,

respectively. The bars represent the mean value of the three averages. The distribution of the full dots indicates for every condition the reproducibility of the results. **b** $H_2O_2$ was added to the sample in (**a**) to induce disulfide bond formation, leading to an increased rate of interactions between the P- and QM-domains. **c** This could be reversed by reducing the disulfide bonds with DTT **d–f** Same as (**a–c**), but with OFS #2 instead of OFS #1. The data were normalized to the HiSiaQM wt/HiSiaP wt interaction (Fig. 6a). A movie of the experiments is shown in Supplementary Movie 2. IFS inward-facing state, OFS outward-facing state, PEG polyethylene glycol, DTT dithiothreitol, TIRF total internal reflection fluorescence. Source data are provided as a Source Data file.

0.64 nM[14]. Thus, if the disulfide bonds in the QM-domain had been formed as intended, VHH$_{QM}$3 should still bind to the HiSiaQM IFS #1/2 constructs but likely not to the OFS #1/2 constructs. Initially, we tried to test this with gelfiltration experiments using the DDM solubilized HiSiaQM constructs. However, the addition of 1 mM $H_2O_2$ seemed not sufficient to lock the HiSiaQM OFS #1/2 constructs, because VHH$_{QM}$3 still bound to the $H_2O_2$-treated transporter (Supplementary Fig. 7). A possible explanation would be that the elevator domain is not sufficiently mobile in DDM and never or at least very rarely visits the OFS conformation. We hence incorporated all four mutants into DOPC-based solid supported bilayers (SSBs) and added 1 mM $H_2O_2$ during the reconstitution procedure to ensure that the disulfide bond was formed as soon as the engineered disulfides contacted each other. We ensured that the $H_2O_2$ concentration did not exceed 1 mM, because we observed aggregation of the QM-domain/VHH complex in the bilayer at $H_2O_2$ concentrations higher than 2 mM.

As previously reported[14], VHH$_{QM}$3 bound strongly to the immobilized wild-type HiSiaQM (Fig. 4a). A mildly reduced level of

interaction was observed for HiSiaQM IFS #1 (Fig. 4b). The HiSiaQM IFS #2 mutant also interacted with VHH$_{QM}$3, but the number of observed binding events was much lower (Fig. 4b). While this might be due to a slightly different conformation of the disulfide linked transporter, we cannot exclude that this observation reflects a non-native state of the IFS #2 construct. As expected, the HiSiaQM OFS #1/2 constructs showed a significantly reduced number of interactions with VHH$_{QM}$3 than the wildtype (Fig. 4c). As expected, only very few interactions with VHH$_{QM}$3 were recorded in the presence of $H_2O_2$ (Fig. 4d). Strikingly, the binding of the nanobody to HiSiaQM OFS #1/2 could be restored, when DTT was added to the experiment (Fig. 4e).

Next, we again used the PEGylation assay[20] to further validate and quantify the disulfide bond formation of HiSiaQM in nanodiscs (Fig. 4f). IFS #1 and OFS #2 clearly showed the expected behavior of a disulfide linked HiSiaQM (compare Fig. 4f with band patterns in Fig. 4g). In contrast, no PEGylation, even after TCEP treatment was observed for IFS #2 (Fig. 4g, h, lanes 11-13). While this might be due to an inaccessibility of the cysteine residues for the PEG reagent, the

reduced ability of this construct to bind VHH$_{QM}$3 led us to exclude this mutant from the experiments shown below. The OFS #1 construct showed the expected behavior in some PEGylation experiments (Supplementary Fig. 8, lanes 5–7), while only one cysteine appeared to be PEGylated in other experiments (Fig. 4g, lanes 5–7).

To verify the structural integrity of the HiSiaQM OFS #1/2 constructs, we performed an experiment where the two proteins were incorporated into the SSB in the absence of sialic acid and $H_2O_2$ but in the presence of the maleimide-AF647 labeled P-domain. The P-domain clearly interacted with the transporter (Fig. 5a, d), probably because some disulfide bonds had spontaneously formed and presumably because of the previously described propensity of the open state HiSiaP to interact with the IFS of HiSiaQM (compare to Fig. 4c). We then added $H_2O_2$ to induce disulfide bond formation and allowed the system to equilibrate for 10 min. Indeed, a significant increase in the observed interactions clearly indicated that the disulfide bonds had been formed (Fig. 5b, e). Next, the sample was supplemented with the reducing agent DTT, and we allowed the system to equilibrate again for 10 min. The number of interactions was now greatly reduced and lower than at the beginning of the experiment, presumably, because after the addition of DTT any preformed disulfide bonds were now broken and most of the HiSiaQM proteins were in the IFS (Fig. 5c, f).

Note that we cannot control the directionality of the incorporation of HiSiaQM into the SSB. Due to the setup of the experiment with the SSB directly on top of the coverslip, we think it unlikely that HiSiaP can interact with upside-down HiSiaQM molecules. Most likely, this fraction of the transporter is silent in the experiment.

Taken together, the experiments above suggested that constructs HiSiaQM IFS #1 and OFS #2 showed the expected behavior in all validation experiments and these constructs were hence considered the best candidates for the TIRF experiments below.

## Interactions of the disulfide-linked P- and QM-domains observed by TIRF microscopy

The single molecule TIRF microscopy setup was then used to investigate the effect of disulfide trapping on the interactions between HiSiaQM and HiSiaP. The first set of experiments was performed with wild-type HiSiaQM, HiSiaQM IFS #1, HiSiaQM OFS #2 and AF647-labeled wild-type HiSiaP in the presence of sialic acid. As expected from previous experiments[14], we observed many interactions between the wild-type P-domain and the wild-type QM-domains (Fig. 6a). A slightly lower level of interactions was observed between the wild-type P-domain and HiSiaQM IFS #1 (Fig. 6b). In contrast, the HiSiaQM OFS #2 mutant showed a significantly lower (−50%) rate of interactions with HiSiaP under these conditions (Fig. 6c)[9,10,20]. Our previous experiments have shown that even in the presence of substrate, the P-domain dynamically opens and closes[8] and the observed interactions might thus occur between HiSiaQM OFS #2 and the fraction of P-domains that visit the open state.

In the next experiment (Fig. 6d), HiSiaQM OFS #2 was combined with wild-type HiSiaP in the absence of sialic acid, conditions, under which the latter does only rarely visit its fully closed state (see above and[8–10]). Strikingly, the open state SBP interacted with HiSiaQM OFS #2 to a similar extent as observed for the two wild-type proteins in the presence of sialic acid (Fig. 6d), in line with the model shown in Fig. 1 (left). Next, we tested the interaction of HiSiaQM IFS #1 and HiSiaP in the absence of sialic acid (Fig. 6e). Here, the level of interactions between the transporter domains and the predominantly open-state SBP was significantly lower compared to the same experiment in the presence of sialic acid (Fig. 6b).

In the next step, HiSiaP CLOSED #2 was added to the mix. As expected, the level of interactions between the locked P-domain and wt HiSiaQM was similar to the experiment with the two wild type proteins (compare Fig. 6a, f). The interactions of HiSiaP CLOSED #2 with HiSiaQM-domains IFS #1/OFS #2 are shown in Fig. 6g, h. Very

clearly, HiSiaP CLOSED #2 preferably interacted with the IFS of HiSiaQM. Taken together, these experiments support the expectation that the closed state of HiSiaP preferentially interacts with the IFS of HiSiaQM and the open state HiSiaP preferentially interacts with the OFS of HiSiaQM.

As mentioned above, the same set of experiments was conducted with the excluded HiSiaQM OFS #1/IFS #2 and HiSiaP CLOSED #1 constructs, which showed some unexpected results during the validation experiments. However, also for these mutants, we recorded very similar results (Supplementary Fig. 9). For the following reasons, this is not unexpected. CLOSED #1 was only excluded because of a missing X-ray structure but behaved as expected in all other experiments (see above). IFS #2 could not be PEGylated in the presence of TCEP, which might simply be due to sterical hindrance. Even if the disulfide bond had not been formed, the IFS appears to be the resting state of TRAP transporters and behaves similar to the wild type in all our experiments. OFS #1 was excluded because the level of disulfide formation was in some experiments lower compared to OFS #2 (Fig. 4, Supplementary Fig. 8), but apparently sufficient in the TRIF experiments.

In the SSB TIRF experiments, we observed a peculiar behavior of the AF647-labeled HiSiaP CLOSED #1/2 mutants (Supplementary Movie 1, 2): A significant fraction of highly mobile SBPs appeared in the focal plane of the microscope and stayed in the membrane plane for a longer time than observed for the wild-type P-domain (Fig. 6a, ref. 14). This effect can be seen in Supplementary Movies 1, 2 and by comparing the projection images in Fig. 6 (bottom row of micrographs) with the single frame images (top row of micrographs). The fast motion leads to the formation of a cloudy background in the projection images. A possible explanation for this observation is that under the corresponding experimental conditions, the three domains of the transporter are in compatible conformational states, i.e., after dissociation, a locked closed P-domain will more quickly find a QM-domain that matches its conformational state, allowing a new interaction to form on the membrane and thus a longer microscopic observation. The significance of these mobile SBPs and their implications for the overall dynamics of the system require further investigation.

## Discussion

The recent advances of the structural knowledge of TRAP transporters have enabled us to engineer locked P- and QM-domains by disulfide engineering[19]. Our high-resolution crystal structure of HiSiaP CLOSED #2, together with ITC, nanoDSF and PEGylation data, demonstrated that our efforts had the intended effect on the P-domain (Figs. 2, 3). The observation that HiSiaP CLOSED #1/2 could be locked in the absence of Neu5Ac was surprising, but the quantification revealed that only a small fraction of the P-domains performs such a closing motion at any point in time. This apo-closed state is very likely short lived[11], otherwise it would have been detected in the single molecule FRET experiments[8]. Future experiments will have to show, whether these rare closure events occur at a similar rate in wild-type HiSiaP (i.e., without engineered cysteines), whether they are spontaneous or rather triggered by small molecules that randomly visit the Neu5Ac binding site. All this considered, we argue that it is still valid to think of apo HiSiaP predominantly adopting the open state in the absence of substrate.

For the QM-domains, the TIRF microscopy- and PEGylation experiments with VHH$_{QM}$3 and apo HiSiaP provided evidence that also here, the engineered disulfide bonds were formed and that the QM-domains still had their native structure (Figs. 4, 5). By combining locked constructs and wild-type proteins in the presence or absence of Neu5Ac, we were able to perform single-molecule TIRF microscopy experiments on different states of the transport cycle (Fig. 6). Our experiments strongly support a key postulation of the current TRAP transporter working hypothesis[14]. That is, the closed state P-domain

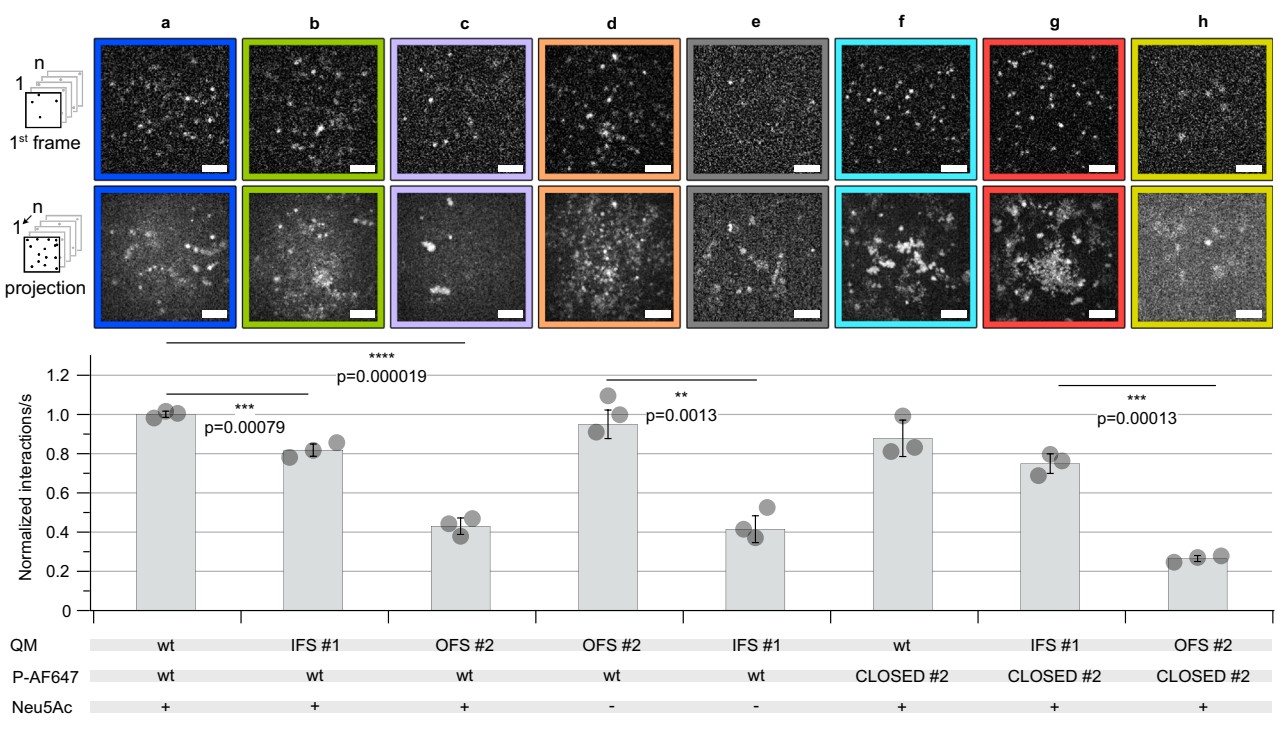

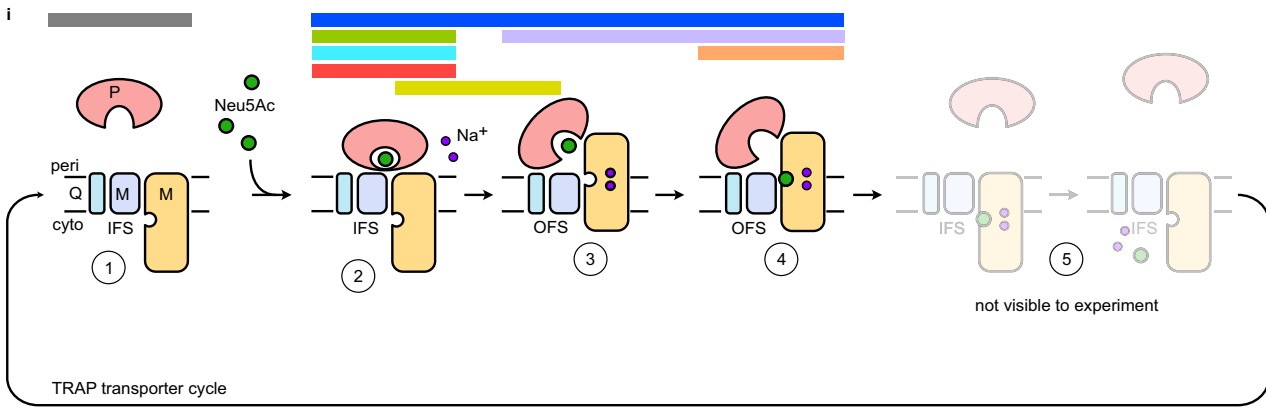

**Fig. 6 | Single molecule TIRF microscopy of trapped TRAP transporter domains. a–h** Top: Visualization of binding of AF647 labeled HiSiaP constructs to the HiSiaQM constructs integrated into DOPC SSBs. The first frames of a typical image sequence with a length of 1000 frames and the corresponding maximum intensity projection of the complete image sequence are shown. Each frame was integrated for 10 ms. Scale bars, 3 μm. A movie showing all experiments can be found in the supplementary information (Supplementary Movie 1). Bottom: Quantification of the imaging data: normalized interactions per second between AF647 labeled HiSiaP constructs and the respective HiSiaQM constructs in the SSB. The data were normalized to the HiSiaQM wt/HiSiaP wt interaction (panel **a**). The statistical significance of differences between selected experiments was assessed by applying a two-sided unpaired Student's *t* test with a 95% confidence interval.

(*$p < 0.01$, **$p < 0.001$). The full dots represent the average results of $n = 3$ independently prepared samples, respectively. The bars represent the mean value of the three averages. The distribution of the full dots indicates for every condition the reproducibility of the results. For each of these three samples, a fresh bilayer was prepared on a new coverslip and $n = 30$ individual measurements were performed. **i** In the scheme, colored horizontal bars mark the probable state of the transporter observed in the experiments with the outline of the same color in (**a–h**). IFS inward-facing state, OFS outward-facing state, PEG polyethylene glycol, DTT dithiothreitol, TIRF total internal reflection fluorescence, SSB solid supported bilayer, DOPC 1,2-dioleoyl-sn-glycero-3-phosphocholine. Source data are provided as a Source Data file.

preferentially interacts with the inward facing elevator and the open state P-domain (i.e., the apo P-domain) preferentially interacts with the outward facing elevator.

Deviating from the transport hypothesis in Fig. 1, the locked P-domain still interacted with the OFS of the transporter to a small degree (Fig. 6h). While a simple structural superimposition of the C-terminal lobe of the closed P-domain onto a model of HiSiaQM in its OFS leads to clashes[14], small conformational rearrangements could render the interaction possible, and an intermediate between states 2

and 3 of the transport cycle would be a candidate to explain the observed state (Fig. 1, Fig. 6i). Another explanation would be just one lobe of the P-domains interacting with one domain (stator or elevator) of the QM-domain, while the other binding interface is not formed. This would presumably result in a lower affinity explaining the smaller number of identified interactions.

Our observation that the engineered disulfide bonds of the HiSiaQM IFS/OFS constructs were readily formed in the DOPC bilayer, indicates that the transporter visits both the IFS/OFS of the transport

cycle, even in the absence of both a Na$^+$-gradient and the P-domain. A similar observation has been made for the GltPh transporter[25,26]. While it is interesting that HiSiaQM appears to behave quite differently in detergent, it is well known that the structure and function of membrane proteins can be strongly influenced by the lipid environment[27]. The role of the lipid environment for elevators was highlighted in a recent MD study on GltPh, which indicated that the switch between the IFS and OFS has a strong influence on the lipid bilayer or vice versa[28].

In previous experiments, where the transporter was reconstituted in liposomes in the absence of both a Na$^+$-gradient and the P-domain, no transport was observed[24]. Thus, simple up and down movement of the transporter is insufficient to drive transport. A possible explanation for this conundrum would be an influence of both the P-domain and the Na$^+$-gradient on the affinity between the QM-domain and its substrate, Neu5Ac. In analogy to VcINDY, sodium ions are most likely involved in Neu5Ac binding. It has recently been shown that the presence of sodium ions stabilizes the substrate binding site of VcINDY[29]. As a reverse conclusion, a lower concentration of sodium ions in the cytoplasm would therefore favor the dissociation of the substrate. The P-domain may further enhance this effect by increasing the local substrate concentration at the periplasmic side of the transporter: The relatively confined space of the HiSiaP-QM complex interface has a volume of $\sim10*10*10$ Å $= 1000$ Å$^3$. In such a small volume, one Neu5Ac molecule would correspond to a very high local concentration in the molar range.

Overall, the results of our study support the current working hypothesis for TRAP transporters and provide experimental evidence that the IFS and OFS of the QM-domains and the closed and open states of the P-domain are indeed conformationally coupled. We show that the HiSiaP CLOSED #1/2, and HiSiaQM IFS/OFS #1/2 proteins are excellent models for mechanistic studies of TRAP transporters. Finally, since all substrate binding proteins share the same architecture, a similar approach may also be useful to further study the mechanism of other SBP-dependent transporters.

## Methods
### Cloning, expression and purification of proteins
The IFS #1/2 and OFS #1/2 constructs were created in a HiSiaQM$_{\Delta Cys}$ backbone (C94A, C325S, C334S, C400S, C485S) and were expressed and purified using a pBADHisTEV vector with N-terminal 10x His-tag and TEV cleavage site[14]. Cysteines were inserted using the method by Liu & Naismith[30]. The membrane domains were expressed in *E. coli* MC1061 cells in LB-medium (100 μg/ml ampicillin) for 2 h at 37 °C. After harvesting the cells at 4000 × *g* for 20 min, the pellets were directly used for purification or stored at −80 °C. For purification, the cell pellets were resuspended in 4 times excess buffer A (50 mM KH$_2$PO$_4$, pH 7.8, 200 mM NaCl and 20% glycerol) and lysed by sonication (40% amplitude, 5 min, pulses 10 s on−5 s off) on ice. The lysed cells were centrifuged for 1 h and 4 °C at 300,000 × *g* and the pellets were resuspended with a homogenizer in 1.5% (w/v) dodecyl-β-D-maltoside (DDM) supplemented buffer A and incubated overnight under gentle shaking at 4 °C. After an identical ultracentrifugation step, the supernatant was mixed with buffer A equilibrated Ni-NTA agarose beads and incubated for 2 h at 4 °C under gentle shaking. Then, the suspension was loaded on a benchtop column, the flowthrough was discarded, and the beads were washed with 100 ml buffer B (50 mM KH$_2$PO$_4$, pH 7.8, 200 mM NaCl and 0.035% DDM) with 22 mM imidazole. 15 ml buffer B with 250 mM imidazole was used to elute the protein from the column. For a final SEC purification step, the protein was concentrated to 500 μl (MWCO 100 kDa) and loaded on an equilibrated Superdex 200 increase 10/300 column with buffer B. The eluted SEC fractions were analysed with SDS-PAGE and HiSiaQM containing fractions were concentrated to around 15 mg/ml, flash-frozen and stored at −80 °C.

The HiSiaP protein was cloned into a pBADHisTEV vector, fusing an N-terminal 6x Histag and a TEV cleavage site to the protein[9].

Mutants were prepared using a QuickChange mutagenesis after Liu & Naismith[30]. *E. coli* BL21 cells were precultured in LB-medium and washed twice with M9 minimal medium. For protein expression, M9 minimal medium was inoculated with bacteria and preincubated 14-16 h at 37 °C before the expression was induced with 500 mg/l L(+)-arabinose. The cells were harvested after 3 h at 37 °C and the pellet was resuspended in 5 times excess of 50 mM Tris (pH 8), 50 mM NaCl. Afterwards the cells were lysed with sonication on ice (40% amplitude, 5 min, pulses 10 s on−5 s off). After centrifugation for 20 min at 75.000 g the supernatant was filtered and supplemented with equilibrated Ni-NTA agarose beads. After binding for 1 h at room temperature, the suspension was loaded on a benchtop column, the flowthrough was discarded and the column was washed with 100 ml resuspension buffer. The protein was eluted with 500 mM imidazole, concentrated to around 5 ml (MWCO 10 kDa) and loaded on an equilibrated HiLoad Superdex 75 16/600 column (50 mM Tris pH 8, 50 mM NaCl). The eluted fractions were collected and checked with an SDS-PAGE. Protein-containing fractions were combined, concentrated and stored after flash-freezing at −80 °C.

The VHH$_{QM}$3 protein was expressed with an N-terminal pelB sequence and a C-terminal His$_6$-tag in *E. coli* WK6 cells[14]. 1 L of TB medium was supplemented with ampicillin (100 μg/ml) and inoculated with a 25 ml preculture of LB-medium. After incubation at 37 °C to a cell density of 0.6, expression was induced with 1 mM IPTG. After overnight incubation at 30 °C, the cells were harvested by centrifugation at 4,000 × *g* for 20 min. and resuspended in 15 ml extraction buffer (200 mM Tris pH 8.0, 0.65 mM EDTA and 500 mM sucrose) for 1 h at 4 °C under gentle shaking. The resuspended cells were treated with 70 ml of 0.25 times diluted extraction buffer and periplasmic lysis was performed overnight at 4 °C. On the next day, the solution was centrifuged at 8000 × *g* for 40 min, filtered (0.45 μm) and mixed with Ni-NTA beads, equilibrated in 0.25 times extraction buffer. The mixture was incubated for 1 h at 4 °C under gentle shaking and transferred to a bench-top column at room temperature. The flowthrough was discarded, the beads were washed with 50 ml wash buffer (50 mM Tris (pH 7.5), 150 mM NaCl and 10 mM imidazole) and the protein was eluted with 10 ml elution buffer (50 mM Tris pH 8, 150 mM NaCl and 500 mM imidazole). For SEC, the protein was concentrated to 5 ml (MWCO 3 kDa) and loaded on an equilibrated HiLoad Superdex 75 16/600 (10 mM Tris pH 7.3 and 140 mM NaCl). The purification steps and eluted fractions were checked with an SDS-PAGE and protein-containing fractions were combined, concentrated and stored after flash-freezing at −80 °C.

### Reconstitution of HiSiaQM constructs in MSP1D1-H5 nanodiscs
The nanodisc reconstitution procedure for the TRAP transporter was based on[14,31]. The purified MSP1D1-H5 and HiSiaQM constructs IFS #1/2, OFS #1/2 were mixed in buffer C (50 mM KH$_2$PO$_4$ (pH 7.8), 200 mM NaCl), supplemented with 50 mM DMPC and 100 mM sodium cholate, to a ratio of 1:60:0.2 (MSP1D1-H5:DMPC:HiSiaQM). The solution was diluted with buffer C to a final concentration of 11 mM DMPC and 22 mM sodium-cholate and incubated for 2 h at 26 °C under gentle shaking. Afterwards, the reconstitution mix was dialysed in a tube with 6−8 kDa MWCO against 500 ml buffer C at 26 °C for16 h with 4 buffer exchanges. Then, the sample was loaded onto a SD 200 3.2/300 column. Peaks were monitored with SDS-PAGE and HiSiaQM as well as MSP1D1-H5 containing fractions were pooled, flash frozen and stored at −80 °C.

### PEGylation assay of HiSiaPQM constructs
The PEGylation assay was carried out according to Mulligan et al.[20]. HiSiaP constructs CLOSED #1/2 (S44C/S171C and S15C/A194C) (11.7 μM) were incubated with either freshly prepared Copper phenanthroline (100 μM) as crosslinking agent or TCEP (0.5 mM) as reduction agent for 1 h on ice. Afterwards, mPEG5K-Maleimide (2 mM) was added

to label any free cysteine and the solution was incubated for 3 h at 23 °C. Samples were visualized on 12% SDS PAGE under nonreductive or reductive conditions and stained with Coomassie blue. Samples including Neu5Ac were prepared in the same fashion, except a preceded incubation of the respective construct with Neu5Ac in a 1:10 molar ratio of concentration for 30 min on ice before adding any agents. HiSiaQM constructs IFS #1/2, OFS #1/2 (6.5 μM) in nanodiscs were incubated with either freshly prepared Copper phenanthroline (100 μM) as crosslinking agent or TCEP (0.5 mM) as reduction agent for 1 h on ice. Subsequently they were incubated in the presence of 0.5% SDS and mPEG5K-Maleimide (2 mM) for 1 h at 23 °C. Samples were visualized on 12% SDS PAGE under nonreductive conditions and stained with Coomassie blue. Images were recorded with a BIORAD ChemiDoc XRS+. Copper phenanthroline was prepared by mixing solutions of 500 mM 1,10-phenanthrolin and 250 mM $CuSO_4$ in a 2:1 ratio.

## Crosslinking time course of HiSiaP CLOSED #1/2

A 200 μL reaction mixture of HiSiaP CLOSED #2 (588 μM), and Copper phenantrolin (100 μM) was set up and incubated on ice. Immediately after adding the Copper phenantrolin 10 μL were taken and given into a quenching solution (500 μM EDTA, 0.5% SDS, 50 mM NaCl, 50 mM Tris, pH 8.0). This was repeated for selected time points. Samples were visualized on 12% SDS PAGE under nonreductive conditions and stained with Coomassie blue. Images were recorded with a BIORAD ChemiDoc XRS + .

## Fluorophore labeling of HiSiaP and VHH$_{QM}$3

The cysteine mutants of HiSiaP (K254C) and VHH$_{QM}$3 (S85C) were expressed and purified as described above[14]. Before labeling, each protein was incubated with 1 mM TCEP (Tris(2-carboxyethyl)phosphine) for 30 min at 4 °C and the reducing agent was removed with a PD Miditrap G-25 column (Cytiva). The eluted protein was directly treated with a 5 times molar excess of AF555 or AF647 maleimide fluorophore (Jena-Bioscience) and incubated for 3 h at 4 °C. Afterwards, the protein was concentrated and washed with a concentrator (MWCO 3 kDa). To remove the remaining unbound label and to check successful labeling, a SEC was performed. The eluted fractions were combined, concentrated and stored after flash-freezing at −80 °C.

For HiSiaP constructs CLOSED #1/2, the labeling procedure was slightly changed to avoid labeling of all three cysteines. First, the protein was incubated with Neu5Ac in 1:10 ratio of concentration for 30 min at room temperature. Then, the protein was incubated with $H_2O_2$ (1 mM per 1 mg/ml protein) for 10 min at room temperature and both compounds, the Neu5Ac and $H_2O_2$ were removed (with PD Miditrap G-25 column (Cytiva)). Afterwards, the protein was labeled with the maleimide fluorophore as described above but without the incubation with a reducing agent.

## ITC experiments

ITC experiments were performed on a MicroCal PEAQ device from Malvern Panalytical. The corresponding software was used for design, measurement and analysis of the ITC experiments. The measuring cell was washed with buffer A and loaded with 120 μM HiSiaP solution in buffer A (50 mM Tris pH 8, 50 mM NaCl). The syringe was loaded with 1.2 mM Neu5Ac, dissolved in the same buffer. For each HiSiaP construct (wildtype, CLOSED #1 and CLOSED #2), the experiments were performed as triplicates.

## nanoDSF experiments

The nanoDSF measurements were performed on a Prometheus NT.48 from NanoTemper using the corresponding software for measurement and analysis. For each measurement, 25 μl of a 1 mg/ml HiSiaP protein solution was prepared and 10 μl was soaked into a glass capillary (standard capillary, NanoTemper). The start and end temperature were set to 20 °C and 90 °C, respectively, and the heating rate to 1 °C/min. The HiSiaP constructs were measured (1) under apo conditions, (2) after 30 min incubation with Neu5Ac at room temperature, (3) after 30 min incubation with Neu5Ac at room temperature and performing a SEC to remove unbound Neu5Ac (see also method preparative SEC runs), and (4) after 30 min incubation with Neu5Ac at room temperature and subsequent 10 min incubation with 1 mM $H_2O_2$ at room temperature and performing a SEC to remove free Neu5Ac and $H_2O_2$.

## X-ray crystallography of HiSiaP

For protein crystallization, HiSiaP was treated with 10 x molar excess of Neu5Ac for 30 min at room temperature and then for 10 min with $H_2O_2$ (1 mM per 1 mg/ml HiSiaP). The solution was passed through a Sephadex G-25 column (Cytiva) and the protein was concentrated to 15 mg/ml for crystallization. Sitting drop crystallization attempts were prepared in a 96 Well 2-Drop MRC crystallization plate using the Crystal Gryphon LCP pipetting robot and 0.2 μl of protein solution and 0.2 μl of crystallization screen. The plates were stored at 20 °C in a Rock Imager 1000 (Formulatrix, US) and imaged automatically. Initial hits were observed in PACT screen condition A12 (0.01 M Zinc chloride, 0.1 M Sodium acetate, pH 5.0, 20% w/v PEG 6000) and several hits in a self-designed optimization screen of this condition with small variations in the conditions. Crystals from one optimized condition were harvested, soaked in mother liquor supplemented with 35% glycerol, flash-frozen and stored in liquid nitrogen. Diffraction data of the crystal was recorded at DESY (Deutsches Elektronen Synchrotron, Hamburg) at beamline P13 (λ = 0.826554 Å) using an EIGER 16 M detector[32] and mxCube v2[33]. The diffraction data were integrated with autoproc[34] and the structure was solved with molecular replacement in PHASER, using the closed-state HiSiaP structure (PDB-ID: 3B50) as model[35]. The structures were refined and modified with PHENIX[36] and COOT[37] and validated with MolProbity[21].

## Solid supported bilayer preparation

Very small unilamellar vesicles (VSUVs) were prepared from a detergent solution according to Grein et al. and Roder et al. [14,38,39]. For the bilayer, a lipid mixture of 31.8 mM DOPC and 0.01 mol% TopFluor-PC (both Avanti Polar Lipids) was prepared in chloroform and dried under a nitrogen stream. The lipids were solubilized in 200 μl HEPES buffer (20 mM HEPES (pH 7.4), 150 mM NaCl) with 40 mM Triton X-100. An aliquot of 20 μl was diluted in 200 μl HEPES buffer. 1 μl of HiSiaQM constructs at a concentration of 9.2 ng/ml were diluted in 500 μl HEPES buffer and the cysteine constructs of HiSiaQM were additionally supplemented with 1 μl of 8 mM $H_2O_2$ and incubated for 10 min at room temperature. Afterwards, 1 μl of this protein solution was added to the lipid mixture. Finally, 200 μl of a 4 mM heptakis(2,6-di-O-methyl)- β-cyclodextrin in water was added and the solution was mixed by vortexing for 2 min. The prepared vesicles were used within 1 h after preparation. 18 ×18 mm coverslips were cleaned overnight in Piranha solution (one-part $H_2O_2$ 30% and two parts concentrated $H_2SO_4$) and rinsed with water. After drying in a nitrogen stream the coverslips were placed into a custom-built chamber with an O-ring as a seal and metal clips to fix the metal insert on top of the coverslip. The bilayers were prepared by adding 400 μl of the VSUVs suspension by filling the well of the sample chamber. Due to electrostatic interactions, a homogeneous bilayer is generally formed after 5 min. Residual vesicles were removed by adding 2 ml of HEPES buffer and removing of only 1 ml of buffer. The chamber was washed 12 times by adding 1 ml HEPES buffer and removing 1 ml. After the last washing step, the final volume in the chamber was 1.4 ml.

## Bilayer binding assay and single molecule imaging

For the VHH$_{QM}$3 staining of HiSiaQM, 2 μL VHH$_{QM}$3-AF555 were added to the surface and incubated for at least 30 min and washed again 5 times as described above by adding 1 ml HEPES buffer and removing 1 ml. The P-domain constructs were diluted 1:20 in HEPES buffer and if necessary for the experiment, the wildtype protein was treated with 10 mM Neu5Ac, incubated for at least 20 min and centrifuged at 14,000 × $g$ for 10 min. Wildtype samples without addition of Neu5Ac and samples which were already incubated with Neu5Ac during labeling were treated the same way. To each bilayer 1 μL of this solution was added. The buffer solution in the sample chamber was mixed carefully by pipetting up and down and incubated for 5 min before measurements were started. Images were acquired at a custom-built, single molecule sensitive, inverted microscope capable of total internal reflection fluorescence (TIRF) microscopy, which was equipped with an sCMOS camera (Prime BSI, Teledyne Photometrics, Tucson, AZ, USA)[38,40]. Illumination with total internal reflection reduced fluorescence excitation to a thin region at the coverslip surface with the benefit of background suppression from fluorescence outside the illuminated region. The illumination beam angle was adjusted by tilting a collimated laser beam in the object focal plane of the imaging lens, until total reflection at the coverslip/medium interface was reached. This was accomplished by moving the laser beam focus laterally in the back focal plane of the objective, respectively in a conjugated plane located outside the microscope. Using a 63× objective lens with a NA 1.45 (Zeiss) resulted in a pixel size of 103 nm. The focus was always carefully adjusted to the TopFluor-PC signal in the bilayer, which was excited by a laser emitting 488 nm (Cobolt 06-MLD, Hübner Photonics GmbH, Kassel, Germany). The focus was stabilized during the measurements by the definite focus system (Zeiss).

For data acquisition, firstly 1000 frames using 640 nm or 561 nm (Cobolt 06, Hübner Photonics GmbH) laser excitation for visualizing the P-Domains labeled by AF647 or the Nb3-AF555, respectively, and then 100 frames using 488 nm were acquired. The exposure time was set to 10 ms and only the central 200 × 200 pixels of the camera chip were read out. For each sample 30 measurements were performed and each experiment was repeated for three independent samples.

Processing of the image sequences was performed in Fiji (version 1.52p)[41]. The 1.000 frames of the P-domain were extracted. Next, the histogram was adjusted to minimum = 0 and maximum = 100 and the background was subtracted using a rolling ball radius of 10. Tracking of single P-domains was performed using the Trackmate plug-in for ImageJ[42]. For spot detection, the LoG-based detector was chosen. The parameter 'estimated blob diameter' was set to 0.75 μm and "sub-pixel localisation" was activated. A threshold of 5 was used for spot filtering. For tracking the "Simple LAP Tracker" was chosen. Gap closing was allowed with a maximum closing distance of 1 μm and a maximum frame gap of two frames. Maximum linking distance was set to 1 μm. Each track was considered as a single interaction of the P-domain with the bilayer.

## Reporting summary

Further information on research design is available in the Nature Portfolio Reporting Summary linked to this article.

## Data availability

The coordinate and diffraction data generated in this study have been deposited in the PDB under accession code 8CP7. The movie data generated in this study are provided as Supplementary Movie files. Source data are provided with this paper as a Source Data file. The coordinate data used in this study are available in the PDB database under accession code 7QE5 and the Alphafold2 predictions of the HiSiaPQM complexes shown in Fig.1 are available in the Supplementary information of[14] [https://doi.org/10.1038/s41467-022-31907-y]. Source data are provided with this paper.

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

## Acknowledgements

G.H. acknowledges funding by the German Research Foundation (DFG), grant number HA 6805/5-1. The synchrotron data was collected at beamline P13 operated by EMBL Hamburg at the PETRA III storage ring (DESY, Hamburg, Germany). We would like to thank Matthias Geyer (Institute of Structural Biology, University of Bonn) for support and discussions.

## Author contributions

G.H. and M.F.P. conceived this study. M.F.P., J.A.R., Y.K., P.H., N.S., J.P.S., G.H.T., U.K. & G.H. planned experiments. M.F.P., J.A.R., Y.K., P.H. and N.S. performed experiments. M.F.P., J.A.R., Y.K., P.H., N.S., J.P.S., G.H.T., U.K. & G.H. analyzed data. G.H. and M.F.P. wrote the paper with input from all authors. G.H. supervised the study and acquired funding.

## Funding

## Competing interests

The authors declare no competing interests.
