## [Peer Review File · Nature Communications]

Reviewers' Comments:

Reviewer #1:

Remarks to the Author:

The manuscript by Peter and colleagues reports studies on the conformational coupling of the sialic acid TRAP transporter HiSiaQM with its substrate binding partner HiSiaP. HiSiaQM is an elevator type transporter with a VcINDY fold. Unlike dimeric VcINDY proteins, HiSiaQM works as a monomer with its scaffold and transport domains residing on two separate polypeptides. How the two domains move relative to each other during transport and, particularly, how the different conformations are coupled to the interaction with the substrate-binding HiSiaP domain is of significant interest. Combining biochemical characterization, X-ray crystallography structure determination and TIRF microscopy, the authors have generated some exciting insights on the HiSiaQMP system. While most of the manuscript is clearly written, the presentation of the TIRF microscopy results needs some clarification. This work should be seen by the readership of Nature Communications.

1. The orientation of the HiSiaQM insertion upon membrane reconstitution is unclear. Did all the transporter molecules insert in a uni-directional way, with their periplasmic surface up as drawn in Fig. 4a-c? If the membrane insertion was random, did the "up-side-down" transporters stay in silence or did such molecules still interact with HiSiaP?

2. The legends for Fig. 5 and Fig. S3 are unclear, making it hard to understand those figures. It is unreasonable to require the reader to dig for additional information in supplementary information to understand a figure.

3. A number of terms are not defined when they first appeared: holo, Neu5Ac, VHHQM3, SSB, ... Without knowing exactly what they are, it is hard for a non-HiSiaQMP person to follow the results.

Minor issues.

4. Do the background HiSiaQMP domains contain any cysteine?

5. P13, Line 26: "Unexpectedly, the locked P-domains still interacted with the OFS of the transporter to a small degree." I would think the locked P-domain has to interact with OFS in order to transfer the substrate to the transporter.

6. The "stator domain" should probably be called the "scaffold domain." There are already too many names for the domain (scaffold, dimerization, trimerization, etc). Even more names will only cause additional confusion.

7. In state 3 in Fig. 1 and the middle illustration of Fig. 5h, the substrate site in the OFS state does not face the extracellular space.

8. For the outward- and inward-facing states, OFS/IFS is used in the text but Co/Ci is used in Fig.

5. This needs to be consistent throughout the manuscript.

Reviewer #2:

Remarks to the Author:

The manuscript by Peter, Ruland, and Kim et al. follows up on two recent manuscripts (one from the same group) describing the cryo-EM structures of TRAP transporters, which are composed of a soluble substrate-binding protein (P-domain) bound to a transmembrane monomeric elevator (QM-domain). This work tests the simple hypothesis that the 'closed' P-domain (substrate-bound) preferentially binds the IFS, and the 'open' P-domain (substrate-free) preferentially binds the OFS. To test this hypothesis, the authors used disulfide engineering to trap both the P-domain and QM domain into open/closed or OFS/IFS, respectively, and tested the various interactions. To my knowledge, this represents the first direct observation of conformational coupling of the P-domain to the QM-domain, which had previously been proposed by the authors and other groups.

The hypothesis is attractive, and the conclusions represent a fascinating transporter mechanism and are of potentially significant interest to the transporter community. However, I have two major concerns. First, crosslinking was not directly validated or quantified, thus making it difficult to

interpret subsequent experiments. Second, the smFRET findings are entirely qualitative and provide limited mechanistic insight. Given the excellent tools previously available and developed in this study, the authors should have been able to provide much more quantitative data and detailed insights into the mechanism and dynamics of the transport cycle. In their current form, the findings appear too preliminary and might be better suited for a more specialized journal.

Essential revisions:

- Much of my concerns regard the crosslinks; both the technical aspects of the experiments establishing the crosslink, and if certain crosslinks are necessary for the conclusions of the paper.
 - a) A major concern with crosslink design of QM is that when I searched the sequence of the wild-type protein, I found five cysteines; yet I was unable to find any discussion of how this is handled. Were these mutated out, or are these present in all constructs? If mutated out, does this construct recapitulate essential features of the protein (clean gel filtration, substrate binding, etc.)? If present, it becomes exceptionally difficult to interpret results without extensive validation because crosslinks, if formed, may be heterogenous or form completely differently as designed. This is especially troublesome for understanding OFS, since as the authors mention, the lack of VHHQM3 binding is not a definitive readout for OFS. It would become necessary to demonstrate that the crosslink forms exclusively with the engineered cysteines; the only way I could see this being done is mapping out disulfides with mass spectrometry.
 - b) For all crosslinks besides CLOSED#2 (crystal structure), the validation is indirect, relative, and qualitative. More quantitative and direct validation is essential to interpret subsequent TIRF experiments.
 - i) The nanoDSF and VHHQM3 binding experiments indirectly suggest crosslink formation but do not give insights into crosslinking efficiency, for reasons described in subsequent points. Direct validation of all crosslinks is necessary to unequivocally prove that most of the protein is crosslinked. This would involve either structure-based approaches (as done with CLOSED #2), mass spec, or demonstrating different cysteine behavior +/- a reducing agent (Fass and Thorpe, Chem Rev 2018). In some fortunate cases, protein may migrate differently in a reducing vs. non-reducing gel (though lack of a difference does not necessarily indicate a lack of crosslinking), which would facilitate quantitation. Another alternative experiment would be quantifying cysteine availability using either direct labeling of free thiols (which assumes disulfides are protected from labeling); or N-ethylmaleimide to block uncrosslinked thiols, followed by disulfide reduction and labeling of resulting free cysteines with a detectable reagent such as PEG-maleimide (which results in a gel shift).
 - ii) Generally, all H₂O₂ experiments may have unexpected effects on protein properties in addition to improvement of crosslinking efficiency, as acknowledged by the authors (“...we observed aggregation of the QM-domain/nanobody complex in the bilayer at H₂O₂ concentrations higher than 2 mM”). These experiments should always be accompanied by testing “reversibility” of the observed effect with reducing agent, as done in Figure 6.
 - iii) P-domain: The crosslinking validation of CLOSED P-domain appears to be nanoDSF experiments combined with addition of H₂O₂, which is proposed to “lock” the P-domain into the closed state after substrate binding. Improvement of T_m certainly suggests likely improvement of crosslinking efficiency, but not absolute crosslinking efficiency nor if the shift in T_m is truly crosslink-dependent (i.e. reversibility with reducing agent). Without direct validation, how do we know if crosslinking efficiency is 8% or 80%? This is particularly relevant for CLOSED #1, which does not have the structural data to bolster confidence. More clarity by direct crosslinking validation would be helpful (as described above).
 - iv) OFS/IFS: The TIRF/VHHQM3 validation of OFS/IFS are excellent experiments to determine relative orientation; however, as currently presented it does not measure crosslinking efficiency. IFS #1 has a similar interaction level as wild type, it is unclear if the crosslink has been established, let alone percent efficiency. Though OFS TIRF data is promising, mutants of transporters commonly shift conformational distributions independent of crosslink formation, and thus crosslinks need to be more rigorously tested (as described above).
 - c) With all this in mind, more clearly defining the rationale and need for CLOSED P-domain compared to wt/substrate P-domain is critical for a reader. As currently presented, I am unsure what insights crosslinking the P-domain is supposed to give, since the authors have previously demonstrated that P protein only visits the closed state in the presence of substrate (pg. 4, lines 18-19). In the smFRET experiments, the crosslinked proteins are tested compared to wt in the

presence of substrate, and the substrate-bound uncrosslinked protein seems to achieve the same thing. More clarity here is needed.

- The key insights of the manuscript come from the single-molecule TIRF analysis; yet the main readout is 'normalized interactions.' This is only qualitatively informative and does not give an adequate sense of how tight the proposed conformational coupling is and how this contextualizes to the transport cycle. Such information, including kinetics and dwell-times, should be extracted from the data. This may lead to more interesting insights than currently presented.
- a) It is surprising that normalized interactions of HiSiaP-open (wt, in absence of substrate) with IFS #1 and #2 are not reported, as relative reduction of this binding compared to OFS would seem to be a critical component of the proposed conformational coupling. Provided that the IFS crosslinks are successful (see above), this should be performed.
- b) The Na⁺-dependence of P-QM binding was not described, but in the provided experimental setup this seems easy to test and important to observe – could sodium coupling be achieved through a different mechanism than VcINDY?
- c) Can crosslink of P-domain be accomplished in absence of substrate? If so, is substrate required for the conformational coupling, or could transient visits to 'apo-closed' bind to IFS? This would be an exceptionally interesting intermediate to capture.
- d) Analysis of dwell-time distributions is important, considering none of the experiments have complete reduction of normalized interactions. What time scales are forming/breaking of interactions happening at, and is there a proposed rate-limiting step? Is there a particular type of interaction event that is conformationally coupled between P and QM? If so, does this give a deeper insight as to the nature of conformationally-coupled binding? The need for such analysis is accentuated by the discussion of the projection images (pg. 13, lines 1-10), which speculates that constrained P-domains can more easily find a QM-domain in a compatible conformational state – quantitation of TIRF data can answer this question.

Minor revisions

Reproducibility and Data Analysis

- For nanoDSF and ITC experiments, average and standard deviation of parameters should be described.
- For ITC experiments, stoichiometry and dH should be provided.
- For nanoDSF experiments, examples of raw data should be provided.
- It would be helpful to see quantification/error in terms of absolute VHHQM3 binding. If VHHQM3 binds tightly, it should be easy to calculate percent binding to QM (and thus, predicted IFS/OFS relative populations) in the TIRF set up.
- Representative SEC profiles of cysteine mutants are necessary to demonstrate that the mutations do not interfere with proper protein folding.

Interpretation and Clarity:

- I don't understand how Supplementary Figure 2 is supposed to clearly demonstrate interaction of VHHQM3 with HiSiaQM OFS#1, since the elution volume compared to HiSiaQM OFS#1 alone is the same. Is the addition of VHHQM3 not expected to significantly change the elution volume, and only change the peak height? Please clarify this in the text.
- It is difficult to tell from the figure schematics when the P-protein is open or closed; I would suggest color-coding this.
- In the context of the proposed mechanism, it is unclear why open P-domain would need to be able to interact with the OFS, yet it appears to do so readily. I would expect the off-rate of this complex to be relatively high, to initiate another round of transport via interaction of another substrate-bound P-domain and IFS. More quantitative data and discussion may add clarity to this.
- CLOSED #2: The crystal structures are the most convincing piece of data proving the crosslink. For the sake of clarity, this figure should be presented first, along with direct validation experiments. As ITC and nanoDSF experiments are only qualitative, indirect measures of crosslink efficiency, these should go in supplement. As a reader, I found it very difficult to follow the current presentation, because ITC and nanoDSF experiments did not instill confidence that the crosslink had actually been formed.
- The authors should be more careful regarding their interpretation of the ITC experiments. Inferring an additional binding site from essentially two injections is unconvincing, especially given

how easily data points can drift due to user modeling and baseline determination. Without the stoichiometry and dH values (which should be provided for all repeats of all experiments, and the avg and SD should be provided in the legend), I cannot assess if these values are realistic given the data. Nevertheless, the authors should be more careful with their discussion (pg. 5, lines 17-25) of pM binding sites (which are indistinguishable from nM sites in ITC), slow koff rates, the fraction of crosslink, and which site is the crosslinked site – as currently presented none of this can be assessed.

- Page 7, lines 26-29 – the alternate density may simply be due to incomplete crosslinking independent of radiation damage since there is currently no direct validation of crosslinking efficiency – almost certainly, some percentage of all engineered proteins do not form the crosslink.
- It is inaccurate to say that an interaction occurs “irreversibly” if non-covalent, in reference to VHHQM3 interaction with QM domain.
- A table describing the predicted cysteine-cysteine distance of all pairs, in both open/closed and OFS/IFS, would be helpful in Supplementary Figure 1.
- What is Figure 6 normalized to, the same as the other figures? This should be stated clearly.
- The interpretation of lower binding of VHHQM3 to IFS#2 because of a “slightly different conformation” (pg. 9, line 1-2) is not substantiated by data; the simpler interpretation is the protein is not in IFS (unhealthy protein, mutant effects on relative IFS population combined with inefficient crosslinking). This should be toned down.

Suggested Experiments

- Strongly suggested: In addition to direct validation, testing VHHQM3 binding to crosslinked QM domains +/- substrate and reducing agent would be good controls. Substrate dependence would validate crosslink rigidity and subsequent experiments +/- substrate (such as Figure 5c-d). Reducing agent dependence would be more likely to fully eliminate the crosslink compared to removing H2O2, further demonstrating that the OFS protein is completely “healthy”. This would also be a good point of comparison to Figure 6, since when the crosslink is broken the protein seems to prefer IFS.
- Optional: If the authors wish to unequivocally determine ITC parameters, I suggest including more titrations with less injection volume (given what I can tell by eye, this should be doable), and/or lowering the concentration of titrant to assess the affinity of the first binding site more accurately. These, combined with titrations +/- reducing agent, would strongly indicate if there are two distinct binding sites and which binding site is the crosslinked site. With all this said, it is sufficient to be more careful with language and interpretation along with direct validation of crosslinks.

Experimental Methods and Detail:

- Please describe the column used in Supplementary Figure 2; based on the retention volumes it is unlikely to be any column described in the methods.
- Buffer conditions of Figure 4 should be described.

Reviewer #3:

None

Response to reviewer comments

We would like to sincerely thank both referees for their time and effort in peer reviewing our manuscript. We feel that your constructive comments have enabled us to greatly improve our study.

Below, our authors response is in red, added text to the manuscript is in blue and the original comments from the reviewers are in black.

Reviewer #1 (Remarks to the Author):

The manuscript by Peter and colleagues reports studies on the conformational coupling of the sialic acid TRAP transporter HiSiaQM with its substrate binding partner HiSiaP. HiSiaQM is an elevator type transporter with a VcINDY fold. Unlike dimeric VcINDY proteins, HiSiaQM works as a monomer with its scaffold and transport domains residing on two separate polypeptides. How the two domains move relative to each other during transport and, particularly, how the different conformations are coupled to the interaction with the substrate-binding HiSiaP domain is of significant interest. Combining biochemical characterization, X-ray crystallography structure determination and TIRF microscopy, the authors have generated some exciting insights on the HiSiaQMP system. While most of the manuscript is clearly written, the presentation of the TIRF microscopy results needs some clarification. This work should be seen by the readership of Nature Communications.

Thank you for this very favourable evaluation of our work!

1. The orientation of the HiSiaQM insertion upon membrane reconstitution is unclear. Did all the transporter molecules insert in a uni-directional way, with their periplasmic surface up as drawn in Fig. 4a-c? If the membrane insertion was random, did the “up-side-down” transporters stay in silence or did such molecules still interact with HiSiaP?

We agree that this should be clarified and we made this clear in the revised version. To the best of our knowledge, HiSiaQM molecules insert in a random manner, with a 50 % fraction of the transporters adopting an orientation where their periplasmic surface faces upward. Based on published structural and functional data (Peter et al 2022, Davies 2023, Mulligan PNAS 2009), the P domain of HiSiaP can only engage in interactions with QM domains that present their periplasmic surface upward. Consequently, transporters that are oriented “upside-down,” with the periplasmic surface facing downward in our SSB setup, cannot participate in interactions with HiSiaP and, therefore, remain silent in terms of their binding activity. We cannot exclude that some P-domain molecules diffuse “underneath” the lipid bilayer, between the lipid bilayer and glass surface, although we consider this a rare event. But for these molecules, the interaction with QM should be similar as for the molecules “above” the glass surface and should not influence our read-out.

p13, l17: “Note that we cannot control the directionality of the incorporation of HiSiaQM into the SSB. Due to the setup of the experiment with the SSB directly on top of the coverslip, we think it unlikely that HiSiaP can interact with upside-down HiSiaQM molecules. Most likely, this fraction of the transporter is silent in the experiment.”

2. The legends for Fig. 5 and Fig. S3 are unclear, making it hard to understand those figures. It is unreasonable to require the reader to dig for additional information in supplementary information to understand a figure.

We have amended the figure and its legend and hope it is clearer now:

Fig. 6: Single molecule TIRF microscopy of trapped TRAP transporter domains. a-h) Top: Visualization of binding of AF647 labelled HiSiaP variants to the indicated HiSiaQM constructs integrated into DOPC SSBs. The first frames of a typical image sequence with a length of 1000 frames and the corresponding maximum intensity projection of the complete image sequence are shown. Each frame was integrated for 10 ms. Scale bars, 3 μm . A movie showing all experiments can be found in the supplementary information (Supplementary Movie 1). Bottom: Quantification of the imaging data: normalized interactions per second between AF647 labelled HiSiaP variants and the respective HiSiaQM constructs in the SSB. The data were normalized to the HiSiaQM wt/HiSiaP wt interaction (panel a)). The statistical significance of differences between selected experiments was assessed by applying a two-sided unpaired Student's t-test with a 95% confidence interval. (* $p < 0.01$, ** $p < 0.001$). The full dots represent the average results of $n = 3$ independently prepared samples, respectively. The bars represent the mean value of the three averages. The distribution of the full dots indicates for every condition the reproducibility of the results. For each of these three samples, a fresh bilayer was prepared on a new coverslip and $n = 30$ individual measurements were performed. i) In the scheme, colored horizontal bars mark the probable state of the transporter observed in the experiments with the outline of the same color in a-h).

3. A number of terms are not defined when they first appeared: holo, Neu5Ac, VHHQM3, SSB, ... Without knowing exactly what they are, it is hard for a non-HiSiaQMP person to follow the results.

Thanks for spotting this. This has been fixed in the revised version.

Minor issues.

4. Do the background HiSiaQMP domains contain any cysteine?

No, they do not. HiSiaP does not contain cysteines and the cysteines in HiSiaQM were removed and the activity of the cys-free construct was checked (now included in the manuscript, see also referee #2). We apologize for not mentioning this in our initial submission.

P5, l5 "The CLOSED #1 and #2 variants of HiSiaP (Fig. 1) were expressed in *E. coli* and were purified analogously to the wild-type protein, which does not have any native cysteines. The proteins behaved like the wild type in gelfiltration runs (Supplementary Fig. 2)."

P10, l14: "Disulfide-linking the elevator domain of HiSiaQM

We expressed and purified the HiSiaQM IFS/OFS double cysteine variants shown in Fig. 1. To avoid any interference of the engineered cysteines with the five native cysteines of HiSiaQM, we replaced the cysteine sidechains with serine or alanine (C94A, C325S, C334S, C400S, C485S), yielding HiSiaQM Δ_{CYS} , which is for simplicity termed HiSiaQM below. According to our established *in vivo* uptake assay^{14,24,25}, the transporter variant was still active, albeit at a reduced level compared to the wild type (Supplementary Fig. 6). In each case, the protein behaved very similar to the wild type during purification (Supplementary Fig. 2)."

5. P13, Line 26: "Unexpectedly, the locked P-domains still interacted with the OFS of the transporter to a small degree." I would think the locked P-domain has to interact with OFS in order to transfer the substrate to the transporter.

This has been clarified in the text. Based on our model, the locked (closed) P-domain interacts with IFS.

p17, l15: "Deviating from the transport hypothesis in Fig. 1, the locked P-domain still interacted with the OFS of the transporter to a small degree (Fig. 6g)."

6. The "stator domain" should probably be called the "scaffold domain." There are already too many names for the domain (scaffold, dimerization, trimerization, etc). Even more names will only cause additional confusion.

We would like to keep it consistent with the preceding paper, but the other names have now been introduced.

p3, l26: "The structures revealed that its four TM helices form a unique helical sheet that wraps around the M-domain and serves to enlarge the stator portion (also known as "scaffold-" or "oligomerization-domain") of the latter."

7. In state 3 in Fig. 1 and the middle illustration of Fig. 5h, the substrate site in the OFS state does not face the extracellular space.

Thanks for spotting this. We amended the figure.

8. For the outward- and inward-facing states, OFS/IFS is used in the text but Co/Ci is used in Fig. 5. This needs to be consistent throughout the manuscript.

Thanks again for spotting this. We amended the figure.

Reviewer #2 (Remarks to the Author):

The manuscript by Peter, Ruland, and Kim et al. follows up on two recent manuscripts (one from the same group) describing the cryo-EM structures of TRAP transporters, which are composed of a soluble substrate-binding protein (P-domain) bound to a transmembrane monomeric elevator (QM-domain). This work tests the simple hypothesis that the 'closed' P-domain (substrate-bound) preferentially binds the IFS, and the 'open' P-domain (substrate-free) preferentially binds the OFS. To test this hypothesis, the authors used disulfide engineering to trap both the P-domain and QM domain into open/closed or OFS/IFS, respectively, and tested the various interactions. To my knowledge, this represents the first direct observation of conformational coupling of the P-domain to the QM-domain, which had previously been proposed by the authors and other groups.

Thank you!

The hypothesis is attractive, and the conclusions represent a fascinating transporter mechanism and are of potentially significant interest to the transporter community. However, I have two major concerns. First, crosslinking was not directly validated or quantified, thus making it difficult to interpret subsequent experiments.

Second, the smFRET findings are entirely qualitative and provide limited mechanistic insight. Given the excellent tools previously available and developed in this study, the authors should have been able to provide much more quantitative data and detailed insights into the mechanism and dynamics of the transport cycle. In their current form, the findings appear too preliminary and might be better suited for a more specialized journal.

Thank you for the constructive comments. We have tried to address these points in our revised manuscript (see below) and hope that our manuscript is now much clearer. Please note that we did not perform any smFRET experiments but rather single molecule TIRF microscopy. Your remark concerning the possible trapping of rare "apo-closed" HiSiaP events was very inspiring, as you will see below and in the revised version of our manuscript.

Essential revisions:

Much of my concerns regard the crosslinks; both the technical aspects of the experiments establishing the crosslink, and if certain crosslinks are necessary for the conclusions of the paper. a) A major concern with crosslink design of QM is that when I searched the sequence of the wild-type protein, I found five cysteines; yet I was unable to find any discussion of how this is handled. Were these mutated out, or are these present in all constructs? If mutated out, does this construct recapitulate essential features of the protein (clean gel filtration, substrate binding, etc.)?

We apologize and we should of course have mentioned that all our data was generated from a HiSiaQM variant that does not have the native cysteines. This was a very unfortunate oversight and we thank the referee for catching this (Referee #1 also noticed this). It is understandable that this missing information has lowered the referee's confidence in our data. This very important information has now been added to the manuscript and we hope the referee now agrees that our data are conclusive, especially considering the additional experiments described below.

P5, 15 "The CLOSED #1 and #2 variants of HiSiaP (Fig. 1) were expressed in *E. coli* and were purified analogously to the wild-type protein, which does not have any native cysteines. The proteins behaved like the wild type in gelfiltration runs (Supplementary Fig. 2)."

P10, 114: "Disulfide-linking the elevator domain of HiSiaQM

We expressed and purified the HiSiaQM IFS/OFS double cysteine variants shown in Fig. 1. To avoid any interference of the engineered cysteines with the five native cysteines of HiSiaQM, we replaced the cysteine sidechains with serine or alanine (C94A, C325S, C334S, C400S, C485S), yielding HiSiaQM_{ΔCys}, which is for simplicity termed HiSiaQM below. According to our established *in vivo* uptake assay^{14,24,25}, the transporter variant was still active, albeit at a reduced level compared to the wild type (Supplementary Fig. 6). In each case, the protein behaved very similar to the wild type during purification (Supplementary Fig. 2)."

To provide further evidence that the Cys-free construct is suited for functional studies, we have performed *in vivo* uptake assays as in Peter et al. 2022 Nat. Comm., as shown below (new Supplementary Fig. 6). While the mutant (blue in figure below) is slightly less active than the wildtype (green in figure below), it clearly represents a working transporter. For an inactive transporter, we would expect a behaviour comparable to the negative control (without TRAP transporter), as already shown for single point mutant in Peter et al. 2022 Nat. Comm.

Supplementary Fig. 6: A sialic acid uptake assay as in SEVY3-based complementation assay^{14,24,25}

In addition, we now show that the gel filtration profiles of the variant without cysteines and of the CLOSED #1/2, IFS #1/2 and OFS#1/2 variants are indeed very similar to that of the wild type transporter.

Supplementary Fig. 2: Gel filtration runs of the different HiSiaP and HiSiaQM constructs used in this study. a) All HiSiaP gelfiltrations were performed on a Superdex 75 16/600 column. **b)** All HiSiaQM gelfiltrations were performed on Superdex 200 10/300 columns but on different FPLC systems. Therefore the runs were aligned on the void peak for easier comparison.

If present, it becomes exceptionally difficult to interpret results without extensive validation because crosslinks, if formed, may be heterogenous or form completely differently as designed. This is especially troublesome for understanding OFS, since as the authors mention, the lack of VHHQM3 binding is not a definitive readout for OFS. It would become necessary to demonstrate that the crosslink forms exclusively with the engineered cysteines; the only way I could see this being done is mapping out disulfides with mass spectrometry.

As explained above, this is not the case and we apologize again for not clearly stating this in our original submission.

b) For all crosslinks besides CLOSED#2 (crystal structure), the validation is indirect, relative, and qualitative. More quantitative and direct validation is essential to interpret subsequent TIRF experiments.

The following points (i-iii) raised by the referee all target the question, whether the engineered disulfides were formed or not. We give short answers to every point of the referee's argumentation and then sum up our evidence for the crosslink formation at the end of this section.

i) The nanoDSF and VHHQM3 binding experiments indirectly suggest crosslink formation but do not give insights into crosslinking efficiency, for reasons described in subsequent points. Direct validation of all crosslinks is necessary to unequivocally prove that most of the protein is crosslinked. This would involve either structure-based approaches (as done with CLOSED #2), mass spec, or demonstrating different cysteine behavior +/- a reducing agent (Fass and Thorpe, Chem Rev 2018).

Please note that we did perform several experiments +/- reducing/oxidation agent (nanoDSF, TIRF in the presence and absence of H₂O₂ and DTT). However, following the referee's suggestions, we provide more supporting experiments now, as discussed in the following points.

In some fortunate cases, protein may migrate differently in a reducing vs. non-reducing gel (though lack of a difference does not necessarily indicate a lack of crosslinking), which would facilitate quantitation. Another alternative experiment would be quantifying cysteine availability using either direct labeling of free thiols (which assumes disulfides are protected from labeling); or N-ethylmaleimide to block uncrosslinked thiols, followed by disulfide reduction and labeling of resulting free cysteines with a detectable reagent such as PEG-maleimide (which results in a gel shift).

Regarding the crosslinking of the P-domain:

- We think that the very similar behaviour of the two CLOSED#1/2 mutants in our biophysical characterization and TIRF experiments together with the X-ray structure of CLOSED#2 is a strong indication that both mutants were disulfide linked.
- Nevertheless, we have further validated the crosslink formation by performing a PEGylation assay as performed by Mulligan et al. 2016. This data is contained in the new Figure 2 and clearly shows the expected result.

Regarding crosslinking of the QM-domain:

We show in our manuscript that in a lipid environment, the OFS mutants do not bind to VHH_{QM3} (hence are clearly different from the IFS) and preferably bind the closed-state P-domain (not expected for a distorted or denatured transporter). This latter point was even shown to be reversible by removing the disulfide bond via addition of DTT. As shown in the next point we have performed additional experiments to show that reduction of the crosslink in the OFS restores binding of VHH_{QM3} (new Figure 4). As requested by the referee, we have now also verified crosslink formation by PEGylation as done

by Mulligan et al. 2016. The new validation experiments are contained in the new Fig. 4, the new Fig. 5 and the new Supplementary Fig. 8. Since the OFS #2 and IFS #1 constructs showed the best match the expected behaviour, we selected these variants for the TIRF experiments in the new Fig. 6 (formerly Fig. 5)

Fig. 4: VHH_{QM3} binding to the IFS/OFS #1/2 variants of HiSiaQM. **a)** The schematic on the left depicts the setup of the single molecule TIRF experiment. The micrograph on the right shows that VHH_{QM3} interacts with the wild-type QM-domains in the bilayer. **b)** Same as a) but with HiSiaQM IFS #1/2. **c)** Same as a), but with HiSiaQM OFS #1/2 and without adding H₂O₂ during the reconstitution. **d)** Same as c) but with adding H₂O₂ during the reconstitution. **e)** Experiment in d) but after the addition of DTT and a 30 min. equilibration time. The scale bars equal 3 μm. **f)** Schematic explaining the PEGylation assay²⁰ for the HiSiaQM variants. **g, h)** Two examples of the PEGylation experiment as described in panel f). Throughout all panels, results that met our expectations are marked with the name of the HiSiaQM construct in green. Validation results that did not meet our expectations are marked in red.

Fig. 5: Following disulfide formation in HiSiaQM OFS #1 in real time. **a)** Top: HiSiaQM OFS #2 was not pre-treated with H₂O₂ and incorporated into DOPC SSBs and interactions with the AF647 labelled P-domain were observed in the absence of sialic acid. The first frames of a typical image sequence with a length of 1000 frames and the corresponding maximum intensity projection of the complete image sequence were shown. Each frame was integrated for 10 ms. Scale bars, 3 μ m. Bottom: Quantification of the imaging data: The statistical significance of differences between selected experiments was assessed by applying a two-sided unpaired Student's t-test with a 95% confidence interval. (*p < 0.01, **p < 0.001). The full dots represent the average results of n = 3 independently prepared samples, respectively. The bars represent the mean value of the three averages. The distribution of the full dots indicates for every condition the reproducibility of the results. **b)** H₂O₂ was added to the sample in a) to induce disulfide bond formation, leading to an increased rate of interactions between the P- and QM-domains. **c)** This could be reversed by reducing the disulfide bonds with DTT **d-f)** Same as a-c, but with OFS #2 instead of OFS #1. The data were normalized to the HiSiaQM wt/HiSiaP wt interaction (Fig 6a). A movie of the experiments is shown in Supplementary Movie 2.

Supplementary Fig. 8: Example of the PEGylation experiment as described in Fig. 4f. Results that met our expectations are marked with the name of the HiSiaQM construct in green. Validation results that did not meet our expectations are marked in red.

ii) Generally, all H_2O_2 experiments may have unexpected effects on protein properties in addition to improvement of crosslinking efficiency, as acknowledged by the authors (“...we observed aggregation of the QM-domain/nanobody complex in the bilayer at H_2O_2 concentrations higher than 2 mM”). These experiments should always be accompanied by testing “reversibility” of the observed effect with reducing agent, as done in Figure 6.

As mentioned in the last point, we show the reversibility of the OFS crosslink by the restoration of VHH_{QM2} binding to the QM domains (new Fig. 4d, e). We think that this is strong evidence against denaturation of the transporter by the H_2O_2 treatment.

iii) P-domain: The crosslinking validation of CLOSED P-domain appears to be nanoDSF experiments combined with addition of H_2O_2 , which is proposed to “lock” the P-domain in its closed state after substrate binding. Improvement of T_m certainly suggests likely improvement of crosslinking efficiency, but not absolute crosslinking efficiency nor if the shift in T_m is truly crosslink-dependent (i.e. reversibility with reducing agent). Without direct validation, how do we know if crosslinking efficiency is 8% or 80%? This is particularly relevant for CLOSED #1, which does not have the structural data to bolster confidence. More clarity by direct crosslinking validation would be helpful (as described above).

See answer to i): Regarding the crosslinking of the P-domain: We think that the very similar behaviour of the two CLOSED#1/2 mutants in our biophysical characterization and TIRF experiments together with the X-ray structure of CLOSED #2 is a strong indication that both mutants were indeed disulfide linked.

We agree that the nanoDSF data show the effect of crosslinking very clearly: For both mutants, in the “bind, lock & wash” experiment, we see an almost complete shift of the melting point by $\sim +10^\circ C$, in the case of CLOSED #2 even stronger than the shift that was observed in the “bind” experiment. We think that it is reasonable to assume that this effect is due to the disulfide bridges stabilizing the closed state. Also, the almost complete shift in T_m very much points towards $>80\%$ disulfide formation, rather than 8 %.

That said, we agree with the referee that the disulfide formation will very likely never be complete and we now mention and discuss this in the manuscript.

p4, l32: “While such crosslinking experiments are very helpful to uncover mechanistically important conformational changes, the presence of a small fraction of non-crosslinked molecules is hard to exclude and should be considered during the interpretation of the results.”

iv) OFS/IFS: The TIRF/VHHQM3 validation of OFS/IFS are excellent experiments to determine relative orientation; however, as currently presented it does not measure crosslinking efficiency. IFS #1 has a similar interaction level as wild type, it is unclear if the crosslink has been established, let alone percent efficiency.

Since our structure of HiSiaQM was released, two other structures of TRAP transporters have appeared and all of these structures show the inward facing state. Thus, this points to the IFS to be the preferred state of the transporter. This is not unexpected and nicely explains the similarity between the wt and IFS TIRF experiments.

Though OFS TIRF data is promising, mutants of transporters commonly shift conformational distributions independent of crosslink formation, and thus crosslinks need to be more rigorously tested (as described above).

As discussed above, we have added many additional supporting experiments, as suggested by the referee.

We would like to summarize this section:

For the P-domain:

We agree with the referee in point i): the determination of a structure (as done for CLOSED #2) is the best approach to show that the disulfide bond can form. Unfortunately, we were not successful in determining the crystal structure of CLOSED #1. Nevertheless, the biophysical characterization (nanoDSF, ITC data (see above)) of both mutants and the TIRF experiments show very comparable results for both mutants, that are clearly different to the observations made with the wildtype protein. Considering all this and the additional proof for the disulfide formation (PEGylation in new Fig. 2), we hope that the referee agrees that there is strong evidence that both P-domain variants are indeed locked in their closed state.

For the QM-domain:

In the first version of our manuscript, we showed a clear dependence of binding events in the presence or absence of reducing and oxidizing reagents for the P-domain. As requested by the referee, we have repeated this experiment with fluorescence-labelled VHH_{QM3} instead of the P-domain as a fluorescence probe. As expected, this experiment shows the opposite effect (new Figs. 4 and 5). We think that the non-binding of VHH_{QM3} to the OFS mutants in combination with their stronger interaction with open-state HiSiaP and the reversibility of the effect is a very strong indication that the transporter was indeed successfully locked in its OFS and maintained its 3D structure.

To provide further prove for the formation of the disulfide bonds, we performed the suggested PEGylation experiment and would like to thank the referee for this suggestion. We used the protocol described by Mulligan et al. 2016 and can clearly see that the introduced Cys residues in OFS#2 and IFS#1 quantitatively formed the desired disulfide bond. For the OFS#1 and IFS#2 variants, the result of the PEGylation assay was less clear and we hence used the former two variants for the discussion and interpretation of the TIRF data in the manuscript.

c) With all this in mind, more clearly defining the rationale and need for CLOSED P-domain compared to wt/substrate P-domain is critical for a reader. As currently presented, I am unsure what insights crosslinking the P-domain is supposed to give, since the authors have previously demonstrated that P protein only visits the closed state in the presence of substrate (pg. 4, lines 18-19). In the smFRET experiments, the crosslinked proteins are tested compared to wt in the presence of substrate, and the substrate-bound un-crosslinked protein seems to achieve the same thing. More clarity here is needed.

Again, please note that we did not perform any smFRET experiments.

The crosslinked P-domain is indeed essential. While we have shown that the P-domain is only stabilized in its closed state in the presence of substrate, we have also previously shown that the P-domain spontaneously opens again, after ~122 ms (Glaenger et al. Biophys J 2017, Peter et al. JMB, 2021). Additionally, our nanoDSF measurements for the wildtype protein from this manuscript show that the protein can release the substrate and adopt an open state. Thus, the crosslink makes sure that this cannot happen or only at a minimal level (if some P-domains are not crosslinked). Therefore, the use of the crosslinked P-domain creates in a significantly better-defined experimental setup.

- The key insights of the manuscript come from the single-molecule TIRF analysis; yet the main readout is 'normalized interactions.' This is only qualitatively informative and does not give an adequate sense of how tight the proposed conformational coupling is and how this contextualizes to the transport cycle. Such information, including kinetics and dwell-times, should be extracted from the data. This may lead to more interesting insights than currently presented.

We appreciate the reviewer's feedback regarding the evaluation of normalised interactions. However, we respectfully disagree with the notion that our analysis is not quantitative.

In our study, we used a rigorous data evaluation to assess the effect of different HiSiaP variants on the binding of HiSiaP to HiSiaQM. We used established experimental protocols and these measurements allowed us to compare the binding profiles of different HiSiaPQM variants and draw conclusions regarding the proposed conformational coupling.

Nevertheless, our analysis still includes a quantitative evaluation of the relative changes in binding between different variants, which nevertheless provides very valuable insights into coupling of HiSiaP and HiSiaQM (as acknowledged by both Referees).

a) It is surprising that normalized interactions of HiSiaP-open (wt, in absence of substrate) with IFS #1 and #2 are not reported, as relative reduction of this binding compared to OFS would seem to be a critical component of the proposed conformational coupling. Provided that the IFS crosslinks are successful (see above), this should be performed.

We appreciate this suggestion. We agree that evaluating the relative reduction in binding compared to OFS is a critical aspect of investigating the proposed conformational coupling. We have included the corresponding data in the new Fig. 6 and the new Supplementary Fig. 9. For the referee's convenience, the data are also summarized here:

b) The Na⁺-dependence of P-QM binding was not described, but in the provided experimental setup this seems easy to test and important to observe – could sodium coupling be achieved through a different mechanism than VcINDY?

Indeed, analysis of the sodium-ion coupling in the transport cycle is very interesting. However, it is impossible to control and manipulate ion gradients within the constraints of our experimental setup based on supported bilayers. This limitation is due to the inherent characteristics of supported bilayers, where the bilayer is not completely closed over the entire coverslip, allowing smaller ions to equilibrate between both sides of the bilayer.

In order to investigate the Na⁺ dependence of P-QM binding and to explore the possibility of alternative mechanisms for sodium coupling, it is necessary to use different membrane model systems that provide greater control over ion gradients. One such system is the use of giant unilamellar vesicles (GUVs), which would also require a completely new experimental procedure. Indeed, we are currently developing such a procedure, however, this is beyond the scope of the current manuscript. Here, we are focused on the interaction between P and QM, rather than on the transport of the substrate (which would require a Na⁺ gradient).

c) Can crosslink of P-domain be accomplished in absence of substrate? If so, is substrate required for the conformational coupling, or could transient visits to ‘apo-closed’ bind to IFS? This would be an exceptionally interesting intermediate to capture.

Thank you very much for this comment, which led us to investigate this more closely. The CLOSED #1 and #2 mutants are indeed an excellent tool to study this. Our experiments revealed that such events are rare but indeed take place. We have added a section about this interesting finding to the manuscript.

p7, l30: “Engineered disulfide links reveal rare closure events of apo HiSiaP

According to the proposed TRAP transporter mechanism (Fig. 1 and ^{3,14}), the state of the SBP in solution is an important factor in the transport cycle. Single molecule FRET experiments had shown that the substrate-bound P-domain reopens after ~ 125 ms ⁸. In contrast, spontaneous closure in the absence of Neu5Ac was not observed experimentally, neither with single molecule FRET nor with pulsed electron paramagnetic resonance (EPR) experiments or X-ray crystallography. ^{8–10} Thus, if at all, such short-lived events must happen quite rarely and are hence difficult to detect. On the other hand, molecular dynamics (MD) simulations of the closely related VcSiaP from *Vibrio cholerae* suggested that “semi-closed” states of the SBP can occur, even in the absence of Neu5Ac¹¹.

Inspired by a referees’ comment, we investigated, whether the disulfide-engineered variants designed in this work can be used as a tool to detect such rare events. In principle, if a particular HiSiaP molecule briefly visits a conformational state that fulfills the geometric requirements for disulfide bond formation, the crosslink should form with a high probability, at least in the presence of an oxidation agent. Given enough time, the growing pool of crosslinked molecules should then be detectable.

Firstly, we investigated the running behavior of the freshly prepared and untreated CLOSED #1/2 mutants on non-reducing SDS-PAGE gels. While CLOSED #1 looked normal, we detected a distinct double band for CLOSED #2 (Fig. 2e, lane 2). This lower band was not present on reducing gels, indicating that the observed band pattern was indeed caused by the spontaneously formed disulfide bond (Fig. 2e, lane 4). Strikingly, in the presence of the oxidation agent copper phenantroline (CuPhe) and notably in the absence of Neu5Ac, the upper band shifted completely to the lower band (Fig. 2e, lane 10). Thus, within the time of the experiment (1h), all HiSiaP CLOSED #2 molecules in the sample had formed the disulfide link. We repeated the experiment with CLOSED #1 and also here, the disulfide bridge was formed and detected by a PEGylation experiment (Fig. 2d, lane 10). The different position of the engineered cysteines along the edge of the substrate binding pocket explains the higher propensity of CLOSED #2 to spontaneously form a disulfide bond, even without an oxidation agent (Supplementary Fig. 5a). This also rationalizes the biphasic ITC and nanoDSF curves of the CLOSED #2 variant (see previous section).

The distinct band shift of the CLOSED #2 variant allowed us to perform a straight-forward time course experiment by determining the fractions of crosslinked and not-crosslinked molecules at different time points by SDS-PAGE analysis (Supplementary Fig. 5bc). The band intensities were quantified and could be fitted with a mono-exponential decay function with a decay constant λ of $\sim 0.01/s$. This results in a half-life ($t_{1/2}$) of ~ 70 s and an average life time τ of 100 s. Assuming that the wild type protein behaves similar to the CLOSED #1/2 variants, we can estimate that on average, the apo HiSiaP protein performs a closing motion every 100 s. Knowing the decay constant λ , we calculated that within one millisecond only about 0.01 % of all P-domains have performed a closing motion. In contrast to the CLOSED #1/2 variants, the wild-type P-domains quickly opens again ¹¹ and thus, only a small fraction of closed state P-domains is present at any time. This explains why such rare events were not detected by experiments ^{8–10}.”

d) Analysis of dwell-time distributions is important, considering none of the experiments have complete reduction of normalized interactions. What time scales are forming/breaking of interactions

happening at, and is there a proposed rate-limiting step? Is there a particular type of interaction event that is conformationally coupled between P and QM? If so, does this give a deeper insight as to the nature of conformationally-coupled binding? The need for such analysis is accentuated by the discussion of the projection images (pg. 13, lines 1-10), which speculates that constrained P-domains can more easily find a QM-domain in a compatible conformational state – quantitation of TIRF data can answer this question.

We appreciate the reviewer's comments to this important topic. However, it is by far not trivial to perform such measurements by TIRF microscopy. They require the acquisition of imaging data at various frame rates and further controls with regard to fluorophore stability, which significantly must exceed the binding durations of the probe molecules. The procedures to analyse binding kinetics by microscopy were recently outlined in detail by the contributions of the Hager and Gebhardt labs (PMIDs 28315485, 32019978 and 33947895). Here, it becomes clear that such analyses required dedicated ways to acquire the data. We acknowledge that the data we acquired so far were not suited for this type of analysis. It would represent a completely new project to analyse the interaction kinetics of the P and QM domains, not to speak of the various mutants. This exceeds by far the scope of the current paper.

Still, we undertook the attempt to analyse the kinetics of the HiSiaPQM variants. We fitted an exponential decay model to the probability distribution of individual interaction times (i.a., binding durations) using OriginPro 8G (OriginLab Corporation, Northampton, MA, USA). In order to assess the possible presence of distinct kinetic steps, the dwell time distributions were modelled using single, double, or triple-modal functions, considering different numbers of populations ($M \in [1-3]$).

$$pdf(A_i; t_i; x) = \sum_{i=1}^M A_i \cdot e^{-x/t_i}$$

(A_i , amplitude; t_i , decay constant)

To identify the optimal model that best describes our data, we calculated the corrected Akaike Information Criterion (AICc) following the approach of Akaike (1974) and Burnham et al. (2002).

$$AIC = n \ln \left(\frac{RSS}{n} \right) + 2k$$

$$AIC_c = AIC + \frac{2k(k+1)}{n-k-1}$$

(n , number of observations; RSS , residual sum of squares; k , number of fitted parameters)

Our analysis revealed that a two-component fit provided the most accurate description of the distribution of interaction times. We found an interaction time of around 30 ms for more than 95% of the interactions, indicating its predominant presence. Additionally, we observed a second step with an interaction time ranging from 200-300 ms, which constituted less than 5% of the total population.

We performed this analysis for each experimental condition and presented the corresponding results in the table below. Notably, our analysis did not uncover any statistically significant differences among the various conditions examined – with two exceptions. The fast binding time of CLOSED #1 and CLOSED #2 (47.1 ± 9 and 42.0 ± 9) were found to be significantly longer than the value for the wt P domain (26.5 ± 8). Indeed, this result is in line with our model. However, we feel that we rather do not want to dwell on the kinetics aspect in view of our initial insight that a meaningful kinetic analysis needs a much broader experimental basis as outlined above.

Furthermore, we think that different experimental methods that allow the direct detection of changes in the mode of action of the transporter, e.g. single-molecule Förster resonance energy transfer (smFRET), should be employed, in order to gain a deeper insight into the conformational coupling of specific interactions between P and QM,

However, smFRET experiments require specialized instrumentation and meticulous labelling of molecules with fluorophores. These factors pose significant challenges and necessitate dedicated resources, expertise, and time to successfully implement and interpret the results. Given the constraints of the present study, incorporating smFRET as an additional experimental setup would extend the scope beyond the current objectives and exceed the available resources.

In view of these conclusions we have changed the statement on page 13 as follows:

P16, I13: “In the SSB TIRF experiments, we observed a peculiar behavior of the AF647-labeled HiSiaP CLOSED #1/2 mutants (Supplementary Movie 1, 2): A significant fraction of highly mobile SBPs appeared in the focal plane of the microscope and stayed in the membrane plane for a longer time than observed for the wild-type P-domain (Fig. 6a and ¹⁴). This effect can be seen in Supplementary Movies 1, 2 and by comparing the projection images in Fig. 6 (bottom row of micrographs) with the single frame images (top row of micrographs). The fast motion leads to the formation of a “cloudy” background in the projection images. A possible explanation for this observation is that under the corresponding experimental conditions, the three domains of the transporter are in “compatible” conformational states, i.e., after dissociation, a locked closed P-domain will more quickly find a QM-domain that matches its conformational state, allowing a new interaction to form on the membrane and thus a longer microscopic observation. The significance of these mobile SBPs and their implications for the overall dynamics of the system require further investigation.”

Table 01: Two-state model interaction time and fraction size for each condition studied. In each condition, the interaction time represents the expected duration of the interaction of HiSiaP with HiSiaQM in the bilayer. The fraction size indicates the proportion of the total population corresponding to a specific interaction time. The reported values are presented as the mean \pm standard deviation.

HiSiaQM	wt	IFS #2	IFS #1	OFS #1	OFS #2
HiSiaP AF647	wt	wt	wt	wt	wt
Neu5Ac	+	+	+	+	+
t_{fast} [ms]	26.5 \pm 8	18.5 \pm 8	18.5 \pm 7	19.5 \pm 8	19.5 \pm 9
F_{fast}	0.98 \pm 1	0.98 \pm 1	0.98 \pm 1	0.97 \pm 1	0.96 \pm 2
t_{slow} [ms]	271 \pm 53	277 \pm 54	283 \pm 55	255 \pm 50	235 \pm 45
F_{slow}	0.03 \pm 1	0.02 \pm 1	0.02 \pm 1	0.03 \pm 1	0.04 \pm 1
HiSiaQM		IFS #2	IFS #1	OFS #1	OFS #2
HiSiaP AF647		CLOSED #1	CLOSED #1	CLOSED #1	CLOSED #1
Neu5Ac		+	+	+	+
t_{fast} [ms]		26.5 \pm 8	30.3 \pm 9	36.2 \pm 14	37.5 \pm 15
F_{fast}		0.98 \pm 1	0.96 \pm 1	0.95 \pm 2	0.95 \pm 2
t_{slow} [ms]		273 \pm 53	231 \pm 41	212 \pm 58	219 \pm 59
F_{slow}		0.03 \pm 1	0.04 \pm 1	0.05 \pm 2	0.05 \pm 2

HiSiaQM	IFS #2	IFS #1	OFS #1	OFS #2
HiSiaP AF647	CLOSED #2	CLOSED #2	CLOSED #2	CLOSED #2
Neu5Ac	+	+	+	+
t_{fast} [ms]	31.6 ± 9	26.9 ± 9	38.7 ± 15	38.9 ± 14
F_{fast}	0.96 ± 1	0.96 ± 1	0.95 ± 2	0.95 ± 2
t_{slow} [ms]	270 ± 52	231 ± 45	223 ± 59	260 ± 57
F_{slow}	0.04 ± 1	0.04 ± 1	0.05 ± 2	0.05 ± 2
HiSiaQM	IFS #2	IFS #1	OFS #1	OFS #2
HiSiaP AF647	wt	wt	wt	wt
Neu5Ac	-	-	-	-
t_{fast} [ms]	29.8 ± 10	28.5 ± 14	37.4 ± 9	32.5 ± 9
F_{fast}	0.95 ± 2	0.95 ± 2	0.95 ± 1	0.95 ± 1
t_{slow} [ms]	242 ± 54	228 ± 49	283 ± 58	231 ± 61
F_{slow}	0.05 ± 2	0.05 ± 2	0.05 ± 1	0.05 ± 1
HiSiaQM	wt	wt		
HiSiaP AF647	CLOSED #1	CLOSED #2		
Neu5Ac	+	+		
t_{fast} [ms]	47.1 ± 9	42.0 ± 9		
F_{fast}	0.96 ± 1	0.95 ± 1		
t_{slow} [ms]	293 ± 62	284 ± 59		
F_{slow}	0.04 ± 1	$0.05 \pm$		

Minor revisions

Reproducibility and Data Analysis

- For nanoDSF and ITC experiments, average and standard deviation of parameters should be described.

See new Fig. 2 and new Supplementary Figs. 3 and 4.

- For ITC experiments, stoichiometry and dH should be provided.

Done. See new Supplementary Fig. 4.

- For nanoDSF experiments, examples of raw data should be provided.

Done. See new Supplementary Fig. 3.

- It would be helpful to see quantification/error in terms of absolute VHHQM3 binding. If VHHQM3 binds tightly, it should be easy to calculate percent binding to QM (and thus, predicted IFS/OFS relative populations) in the TIRF set up.

Unfortunately, there are certain limitations that prevent us from calculating percent binding to the QM domain and predicting the relative populations of IFS/OFS in the SSB. One challenge arises from the lack of knowledge regarding the absolute surface density of the QM domain in the SSB. Without this information, it is not possible to reliably determine the binding ratio of the nanobody. Furthermore, when the QM domain is added to the SSB, the concentration becomes presumably quite high resulting in QM domains in a large proximity far below the limit of optical resolution. As a result, after introducing VHHQM3, the bound QM domains cannot be resolved individually due to the resolution limit. Consequently, it is not possible to accurately determine the number of bound VHHQM3.

- Representative SEC profiles of cysteine mutants are necessary to demonstrate that the mutations do not interfere with proper protein folding.

Done, see above and new Supplementary Fig. 2.

Interpretation and Clarity:

- I don't understand how Supplementary Figure 2 is supposed to clearly demonstrate interaction of VHHQM3 with HiSiaQM OFS#1, since the elution volume compared to HiSiaQM OFS#1 alone is the same. Is the addition of VHHQM3 not expected to significantly change the elution volume, and only change the peak height? Please clarify this in the text.

Because of the small size of the nanobody compared to the transporter in a detergent micelle, the shift is indeed too small.

- It is difficult to tell from the figure schematics when the P-protein is open or closed; I would suggest color-coding this.

We have added more labels to the schematics to clarify this. We would like to avoid adding yet another color.

- In the context of the proposed mechanism, it is unclear why open P-domain would need to be able to interact with the OFS, yet it appears to do so readily. I would expect the off-rate of this complex to be relatively high, to initiate another round of transport via interaction of another substrate-bound P-domain and IFS. More quantitative data and discussion may add clarity to this.

As discussed above, we cannot determine reliable off-rates with our setup.

- CLOSED #2: The crystal structures are the most convincing piece of data proving the crosslink. For the sake of clarity, this figure should be presented first, along with direct validation experiments. As ITC and nanoDSF experiments are only qualitative, indirect measures of crosslink efficiency, these should go in supplement. As a reader, I found it very difficult to follow the current presentation,

because ITC and nanoDSF experiments did not instill confidence that the crosslink had actually been formed.

Note that we did not use ITC to quantify the crosslinking. We used the method to check that the HiSiaP variants still bound to Neu5Ac. Upon thinking about the referees' suggestion we decided to move the data to the SI (new Supplementary Fig. 3) and combined the nanoDSF data with the PEGylation assay (new Fig. 2).

- The authors should be more careful regarding their interpretation of the ITC experiments. Inferring an additional binding site from essentially two injections is unconvincing, especially given how easily data points can drift due to user modeling and baseline determination. Without the stoichiometry and dH values (which should be provided for all repeats of all experiments, and the avg and SD should be provided in the legend), I cannot assess if these values are realistic given the data. Nevertheless, the authors should be more careful with their discussion (pg. 5, lines 17-25) of μM binding sites (which are indistinguishable from nM sites in ITC), slow k_{off} rates, the fraction of crosslink, and which site is the crosslinked site – as currently presented none of this can be assessed.

Thanks for this comment. This is now discussed much more carefully.

p5, l26: " The second variant, HiSiaP CLOSED #2, showed a peculiar behavior in the ITC experiments that was distinct from any other HiSiaP variant that we had previously tested. The titration curves of HiSiaP CLOSED #2 consistently had a biphasic shape and could not be fitted with a simple 1:1 binding model (Supplementary Fig. 3g-i). Instead, a model with two independent binding reactions was used. One of the reactions was fully resolved and the average K_D was determined to be in the micromolar range ($\sim 2 \mu M$). The other reaction appeared to have a higher affinity but was not fully resolved in our titration experiments and hence its thermodynamic parameters could not be reliably determined. We speculate that this observation was due to a fraction of HiSiaP CLOSED #2 having formed its engineered disulfide bond. Overall, our ITC data suggested that the presence of the cysteines alone, in the absence of an oxidizing agent, results in only a small fraction of the HiSiaP with the desired disulfide bonds formed."

- Page 7, lines 26-29 – the alternate density may simply be due to incomplete crosslinking independent of radiation damage since there is currently no direct validation of crosslinking efficiency – almost certainly, some percentage of all engineered proteins do not form the crosslink.

We have added this point to the manuscript.

p9, l21: "We found that traces of a second conformation of C194 were visible in the electron density (Fig. 3c).²² This might either indicate incomplete formation of the disulfide bridge or radiation induced reduction of the latter²³."

- It is inaccurate to say that an interaction occurs "irreversibly" if non-covalent, in reference to VHHQM3 interaction with QM domain.

Amended.

- A table describing the predicted cysteine-cysteine distance of all pairs, in both open/closed and OFS/IFS, would be helpful in Supplementary Figure 1.

We have added such a table. New Supplementary Table 1.

Supplementary Table 1: Details of the engineered disulfide bridges

Construct	C _β distance (Å)	Based on model
CLOSED #1 (HiSiaP S44C-S171C)	4.8	3B50 ⁶
CLOSED #2 (HiSiaP S15C-A194C)	4.0	3B50 ⁶
IFS #1 (HiSiaQM _{ΔCys} A492C-Q539C)	5.9	7QE5 ¹⁴
IFS #2 (HiSiaQM _{ΔCys} M297C-M259C)	4.8	7QE5 ¹⁴
OFS #1 (HiSiaQM _{ΔCys} A492C-L512C)	6.1	Model of OFS ¹⁴
OFS #2 (HiSiaQM _{ΔCys} M297C-Y252C)	5.0	Model of OFS ¹⁴

- What is Figure 6 normalized to, the same as the other figures? This should be stated clearly.

The information has been added. The data were always normalized to the experiment with the wild type proteins.

- The interpretation of lower binding of VHHQM3 to IFS#2 because of a “slightly different conformation” (pg. 9, line 1-2) is not substantiated by data; the simpler interpretation is the protein is not in IFS (unhealthy protein, mutant effects on relative IFS population combined with inefficient crosslinking). This should be toned down.

We have added this point and toned the statement down:

p11, l7: “While this might be due to a slightly different conformation of the disulfide linked transporter, we cannot exclude that this observation reflects a non-native state of the IFS #2 variant.”

Suggested Experiments

- Strongly suggested: In addition to direct validation,

testing VHHQM3 binding to crosslinked QM domains +/- substrate and reducing agent would be good controls.

Done, see above and new Fig. 4.

Substrate dependence would validate crosslink rigidity and subsequent experiments +/- substrate (such as Figure 5c-d). Reducing agent dependence would be more likely to fully eliminate the crosslink compared to removing H₂O₂, further demonstrating that the OFS protein is completely “healthy”. This would also be a good point of comparison to Figure 6, since when the crosslink is broken the protein seems to prefer IFS.

Unfortunately, this comment is not clear to us, since we did actually use DTT to remove the crosslink as shown in new Fig. 5 (former Fig. 6).

- Optional: If the authors wish to unequivocally determine ITC parameters, I suggest including more titrations with less injection volume (given what I can tell by eye, this should be doable), and/or lowering the concentration of titrant to assess the affinity of the first binding site more accurately. These, combined with titrations +/- reducing agent, would strongly indicate if there are two distinct binding sites and which binding site is the crosslinked site.

As mentioned above, ITC was primarily used to show that the CLOSED#1 and #2 mutants still bind Neu5Ac and can hence adopt the closed state this has now been clearly explained in the manuscript. As stated above, this has now been worded more carefully.

With all this said, it is sufficient to be more careful with language and interpretation along with direct validation of crosslinks.

Thank you again for your very thorough analysis of our manuscript. We think that the thought-provoking points you raised allowed us to significantly improve the manuscript and led us to discover that spurious closing events of apo HiSiaP do indeed happen.

Experimental Methods and Detail:

- Please describe the column used in Supplementary Figure 2; based on the retention volumes it is unlikely to be any column described in the methods.

Done. It was a Superose Increase 6 3.2/300 column.

- Buffer conditions of Figure 4 should be described.

Also here the HEPES buffer (20 mM HEPES (pH 7.4), 150 mM NaCl) was used.

Reviewers' Comments:

Reviewer #1:

Remarks to the Author:

The authors have significantly improved the clarity of the manuscript, and I have only some minor suggestions.

1. Line 106: define Neu5Ac.
2. Various disulfide mutants, CLOSED, OFS and IFS, are all called "variants." This is confusing, particularly for a paper in microbiology. As such mutants are designed and engineered, they should be called constructs.
3. Table S1: Cell dimensions -> Unit cell dimensions.
4. Proper figure legends are missing for Figs. S3, S6.
5. Fig 1 legend: Reference #13 is cited as the source of the reaction cycle shown in the figure. I was unable to find such a description in reference #13.

Reviewer #2:

Remarks to the Author:

I appreciate the authors' thorough and detailed responses to the critiques. The manuscript has vastly improved with the justifications of Cys-free constructs, key validations for crosslinks, and re-focusing the results/discussion on validated OFS#2 and IFS#1. I concede misunderstanding the TIRF setup and appreciate the authors' explanation regarding the limitations of calculating kinetics. I would be interested in clarification of the following points:

- Difficulties of comparative analysis notwithstanding, is there any significance or speculation to the consistent requirement of two-component fits to the TIRF data?
- After initially reviewing the manuscript, I was interested in knowing if analysis of photobleaching in the TIRF data (i.e. SiMPull) was possible, and could distinguish between the monomeric complex predicted by previous cryo-EM structures, or potential dimerization recently reported by Currie & Davies et al (<https://doi.org/10.1101/2023.08.28.549404>).

Otherwise, I congratulate the authors and recommend this manuscript for publication.

We would like to thank both referees again for their time and effort in reviewing our manuscript! We are very happy about your positive comments on our revised version.

REVIEWERS' COMMENTS

Reviewer #1 (Remarks to the Author):

The authors have significantly improved the clarity of the manuscript, and I have only some minor suggestions.

1. Line 106: define Neu5Ac.

Done.

"The first structural information about TRAP transporters was provided by two high-resolution crystal structures of the P-domain of the sialic acid (**Neu5Ac, N-Acetylneuraminic acid**) TRAP transporter HiSiaPQM from *Haemophilus influenzae* in both the apo and holo states (i.e., substrate-bound and -free, respectively) 5,6."

2. Various disulfide mutants, CLOSED, OFS and IFS, are all called "variants." This is confusing, particularly for a paper in microbiology. As such mutants are designed and engineered, they should be called constructs.

Done.

3. Table S1: Cell dimensions -> Unit cell dimensions.

Done.

4. Proper figure legends are missing for Figs. S3, S6.

Done.

Supplementary Fig. 3 | Isothermal titration calorimetry runs for the indicated HiSiaP constructs. In each panel, the raw data are shown on the top and the integrated heat signals, as well as the thermodynamic parameters at the bottom. Source data are provided as a Source Data file.

Supplementary Fig. 6 | A sialic acid uptake assay as in SEVY3-based complementation assay 14,24,25. Briefly, *E. coli* cells without a functioning sialic acid uptake system were supplemented with either wild-type HiSiaPQM, the HiSiaPQM Δ Cys mutant or an empty plasmid. Growth of these bacteria was monitored in minimal media with sialic acid as the sole source of carbon and energy. Data of n=3 independent experiments are presented as mean values (colored circles) +/- standard deviation (error bars). Source data are provided as a Source Data file.

5. Fig 1 legend: Reference #13 is cited as the source of the reaction cycle shown in the figure. I was unable to find such a description in reference #13.

Thank you for catching this mistake!

Reviewer #2 (Remarks to the Author):

I appreciate the authors' thorough and detailed responses to the critiques. The manuscript has vastly improved with the justifications of Cys-free constructs, key validations for crosslinks, and re-focusing the results/discussion on validated OFS#2 and IFS#1. I concede misunderstanding the TIRF setup and appreciate the authors' explanation regarding the limitations of calculating kinetics. I would be interested in clarification of the following points:

- Difficulties of comparative analysis notwithstanding, is there any significance or speculation to the consistent requirement of two-component fits to the TIRF data?

A possible explanation would be that one component describes the actual interaction, while the other component accounts for the bleaching of the dyes. To really get to the bottom of this, more experiments with different time intervals and tightly controlled oxygen concentrations would be needed.

- After initially reviewing the manuscript, I was interested in knowing if analysis of photobleaching in the TIRF data (i.e. SiMPull) was possible, and could distinguish between the monomeric complex predicted by previous cryo-EM structures, or potential dimerization recently reported by Currie & Davies et al (<https://doi.org/10.1101/2023.08.28.549404>).

This is a very interesting suggestion and we plan to perform such experiments in the future.

Otherwise, I congratulate the authors and recommend this manuscript for publication.